# Recent autumn sea ice loss in the eastern Arctic enhanced by summer Asian-Pacific Oscillation

Botao Zhou [1,2] ✉, Ziyi Song [1,2], Zhicong Yin [1,2], Xinping Xu[1,2], Bo Sun[1,2], Pangchi Hsu[1,2] & Haishan Chen [1,2]

Recent rapid Arctic sea ice loss was documented as combined results from anthropogenic forcing and climate system internal variability. However, the role of internal variability is not well understood. Here, we propose that the Asian-Pacific Oscillation (APO), an intrinsic atmospheric mode featuring out-of-phase variations in upper-tropospheric temperatures between Asia and the North Pacific, is one driver for autumn sea ice variability in the eastern Arctic. The positive summer APO favors warming of the mid-latitude North Atlantic sea surface temperatures. This warming persists to autumn and in turn triggers strong anticyclonic anomalies over the Barents-Kara-Laptev Seas and weak lower-tropospheric cyclonic anomalies over the East Siberian Sea, enhancing moisture transport into the eastern Arctic. Such changes consequently increase lower-tropospheric humidity, downwelling longwave radiation, and surface air temperature in the eastern Arctic, thereby melting sea ice. Hence, a recent tendency of the summer APO towards the positive phase accelerates autumn sea ice loss in the eastern Arctic.

In the context of global warming, the Arctic has warmed faster than global average by a factor of 2 or more since the mid-20th century[1,2]. Under such a background, the Arctic sea ice has melted at an accelerating pace particularly in autumn during recent decades[3]. A number of studies have documented impacts of Arctic sea ice loss on local and remote climate[4–15]. For instance, autumn sea ice loss in the Barents-Kara Seas tends to increase the frequency of cold winters and snow storms in Eurasia[16–20] via influencing the Eurasian blocking[21,22], the East Asian winter monsoon circulations[23,24], the land process[25] and so on. The decline of autumn sea ice in the Kara-Laptev Seas even affects the subsequent summer precipitation over East Asia through the seasonal persistence of snow depth and soil moisture[26]. The sea ice reduction in the Chukchi-East Siberian Sea is inclined to cool temperatures in central North America[27] and that around Greenland is closely linked to the variability of surface temperatures in northern Europe and eastern North America[28,29]. Nevertheless, some studies have reservations about the effects of Arctic sea ice loss[30–32]. For example, Refs. 33,34 reported a weak response of the Eurasian winter climate to the reduction of Arctic sea ice.

Although no consensus is reached for the climate effects of sea ice loss in the Arctic, the physical drivers for Arctic sea ice loss have sparked increasing interest. It is indicated that the secular trend of Arctic sea ice loss is attributed to anthropogenic forcing[1,35]. The internal variability within the climate system also plays important roles[36–42]. For example, Ref. 37 revealed that anomalous anticyclonic circulation over Greenland and the Arctic Ocean in summertime may contribute as much as 60% to the September Arctic sea ice loss since 1979. Some atmospheric modes of internal variability, such as the Arctic Dipole[43], the Arctic Oscillation/North Atlantic Oscillation[44–46], and the Pacific North American pattern[47], have also been highlighted as physical drivers for the Arctic sea ice variability.

[1]Collaborative Innovation Center on Forecast and Evaluation of Meteorological Disasters/Key Laboratory of Meteorological Disaster, Ministry of Education/ Joint International Research Laboratory of Climate and Environment Change, Nanjing University of Information Science and Technology, Nanjing, China. [2]School of Atmospheric Sciences, Nanjing University of Information Science and Technology, Nanjing, China. ✉e-mail: zhoubt@nuist.edu.cn

The Asian-Pacific Oscillation (APO)[48], featuring a seesaw change in the upper-tropospheric temperatures between Asia and the North Pacific, is an atmospheric mode of internal variability over the extratropical Asian-Pacific sector and most dominant in the summertime[49]. When the troposphere is warming over the Asian continent, the troposphere is cooling over the North Pacific, and vice versa. In essence, the APO pattern reflects the variability of zonal thermal contrast between Asia and the North Pacific. Due to its hemispheric nature, summer APO exerts prominent effects on large-scale atmospheric circulations and climate in and beyond the Asian-North Pacific sector[48–54]. An emerging question is whether it also plays a role in the variability of Arctic sea ice. In this study, we are motivated to examine their linkage based on the reanalysis data including the HadISST1 sea ice concentration (SIC), the ERSST v5 sea surface temperature (SST), and the NCEP/NCAR atmospheric reanalysis as well as the Community Earth System Model Large Ensemble (CESM-LE) simulation data (see "Data and Methods" section for details), and demonstrate that summer APO may act as a driver of autumn sea ice variability in the eastern Arctic.

## Results

### Observed relationship between summer APO and autumn sea ice in the eastern Arctic

We first examine the co-variability of summer (June-July-August, JJA) upper-tropospheric (500-200 hPa) eddy temperature (UTT, defined as the departure of temperature from its zonal mean) over the Asian-Pacific sector (25°–65°N, 75°E-120°W) and autumn (September-October-November, SON) SIC in the eastern Arctic (70°–83°N, 30°E-180°) by the maximum covariance analysis (MCA). The spatial patterns associated with the first leading MCA mode, accounting for 62.9% of the squared covariance between the two fields analyzed, characterize that positive UTT anomalies over Asia and negative UTT anomalies over the North Pacific (Fig. 1a) during summer are coupled with autumn sea ice decline in the eastern Arctic (Fig. 1b). The correlation coefficients of their corresponding temporal expansion series, exhibiting consistent interannual and interdecadal variations over the record (Fig. 1c), are significantly correlated at $r = 0.62$ ($p < 0.01$). A similar result can be generally obtained if we redo the analysis over the satellite record period (1980–2019) directly using the satellite data of SIC (Supplementary Fig. 1).

Of note, the UTT spatial distribution in the leading MCA mode (Fig. 1a) highly resembles its first leading empirical orthogonal function (EOF1) pattern, featured by a zonal seesaw UTT oscillation over the Asian-Pacific sector (Supplementary Fig. 2a). The EOF1 mode, explaining 31% of the total variance, denotes the dominant pattern of the APO variability[48]. Here, we define the time series of the leading MCA of UTT (red line in Fig. 1c) as the APO index for observational analysis. This index is highly correlated with the principal component (PC1) series of the EOF1 pattern (Supplementary Fig. 2b, $r = 0.98$, $p < 0.01$). Such a high correlation suggests that the index we define can measure the variability of summer APO. Similar to the previous studies[53–55] which reported decadal variations, the summer APO over the course of 1950–2019 exhibits decadal variability with the positive phase from the 1950s to the mid-1960s as well as after the mid-2000s and the negative phase in between (Fig. 1c and Supplementary Fig. 2b). Similarly, the temporal expansion series of the leading MCA for autumn SIC (blue line in Fig. 1c) is employed as the SIC index (SICI). This index is also strongly correlated with the area-weighted mean of autumn SIC over the eastern Arctic ($r = -0.98$, $p < 0.01$), indicating that it can represent autumn sea ice variability in the eastern Arctic. A positive SICI value signifies a lower-than-normal SIC in the eastern Arctic, and vice versa. The area-weighted mean of autumn SIC over the eastern Arctic and the summer APO are correlated at $r = -0.59$ ($p < 0.01$).

The close connection of summer APO with the autumn sea ice variability in the eastern Arctic is also supported by the regression of autumn SIC anomalies (relative to the climatology of 1950–2019) against the summer APO index. As shown in Fig. 1d, a positive phase of summer APO (i.e., warmer UTT over Asia accompanied by cooler UTT over the North Pacific) well corresponds to a uniform reduction of sea ice in the eastern Arctic during the following autumn. The squared wavelet coherence analysis for summer APO and autumn SIC indices (Supplementary Fig. 3) further illustrates that they show a predominant in-phase coherency in 5–8 year, 8–12 year, and 16–24 year bands. Therefore, we hypothesize that summer APO may act as a precursor driving eastern Arctic sea ice variability in autumn on interannual-to-decadal time scales. When summer APO is in the positive phase, the eastern Arctic sea ice tends to decline in the subsequent autumn. Conversely, if summer APO lies in the negative phase, the eastern Arctic is inclined to

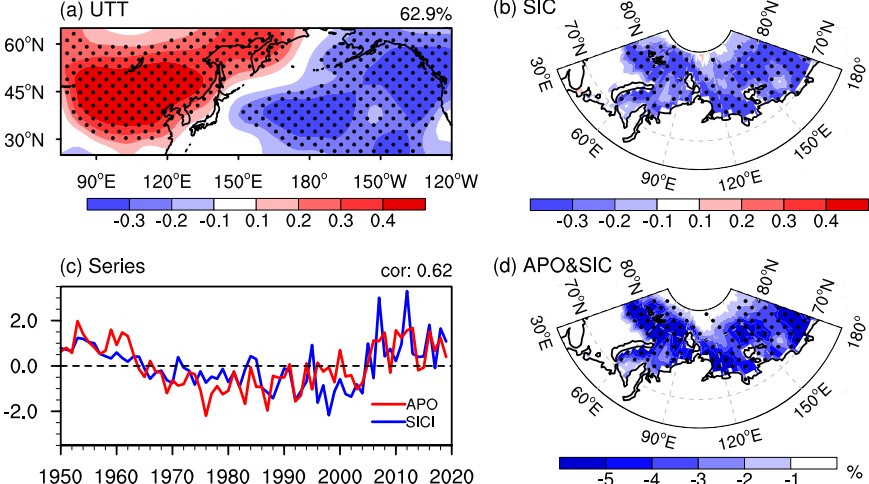

**Fig. 1 | Coupled patterns between summer upper-tropospheric (500-200 hPa) eddy temperature (UTT) over the Asian-Pacific sector and autumn sea ice concentration (SIC) in the eastern Arctic. a–b** Spatial patterns of the first leading mode of the maximum covariance analysis for (**a**) summer UTT over the Asian-Pacific sector and (**b**) autumn SIC in the eastern Arctic over the period 1950–2019. **c** Time series of the UTT (red line, defined as the Asian-Pacific Oscillation (APO)) index) and SIC (blue line, defined as the SIC index (SICI)) patterns shown in (**a**) and (**b**), respectively. **d** Regression of autumn SIC anomaly (%) with the normalized summer APO index over the period 1950–2019. Areas significant above the 90% confidence level are dotted in (**a**), (**b**), and (**d**). This figure was created by the NCAR Command Language (NCL) Version 6.6.2 (https://www.ncl.ucar.edu/). Source data are provided as a Source Data file.

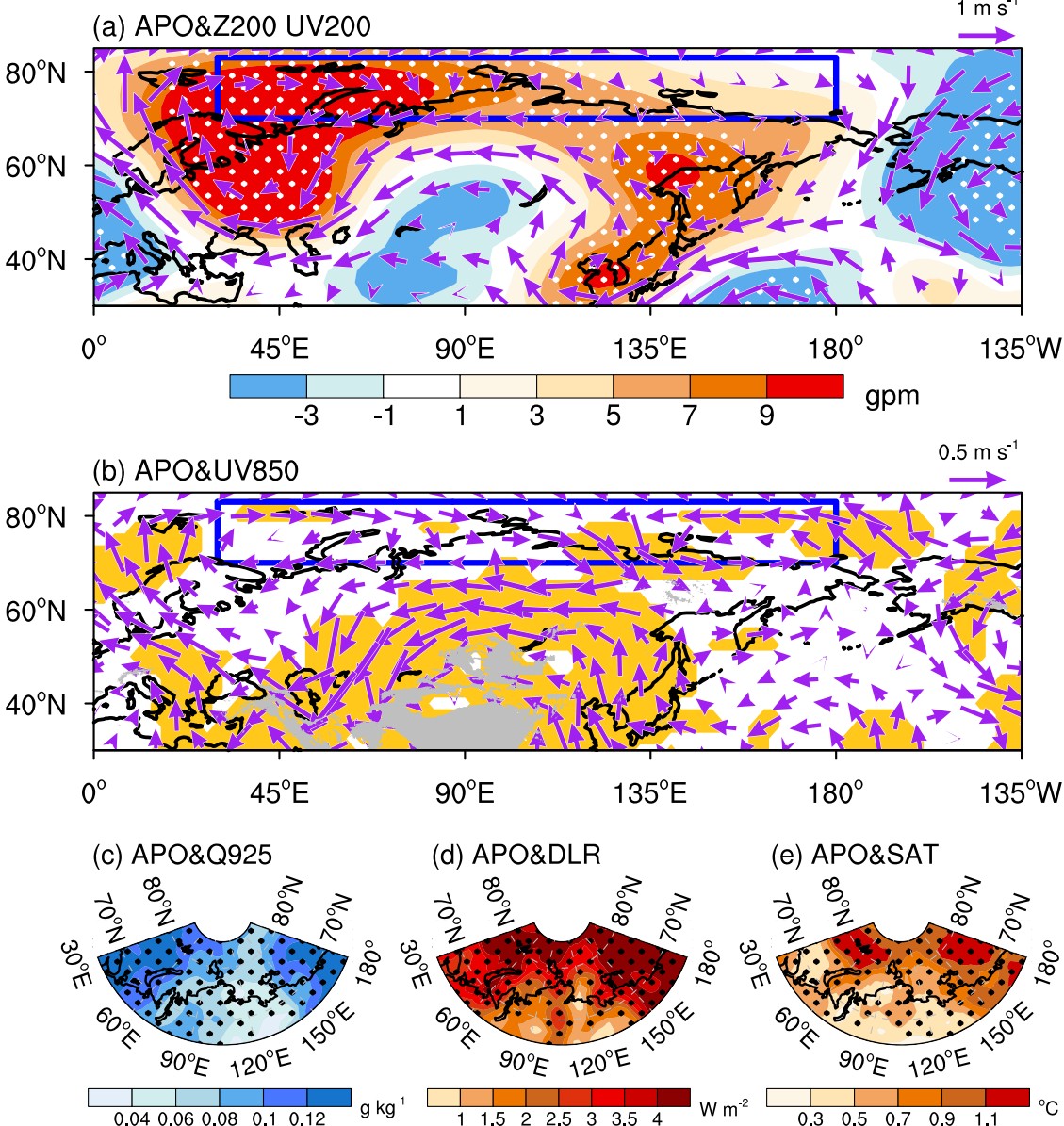

**Fig. 2 | Asian-Pacific Oscillation (APO)-regressed autumn anomalies in atmospheric circulations and thermodynamic processes. a** 200 hPa horizontal wind (UV200, vectors, m s$^{-1}$) and geopotential height (Z200, shading, gpm), **b** 850 hPa horizontal wind (UV850, vectors, m s$^{-1}$), **c** 925 hPa specific humidity (Q925, g kg$^{-1}$), **d** downwelling longwave radiation (DLR, W m$^{-2}$), and **e** surface air temperature (SAT, °C) in autumn regressed onto the normalized summer APO index over the period 1950–2019. Areas significant above the 90% confidence level are shaded in (**b**) and dotted in other panels. The blue box in (**a**) and (**b**) outlines the eastern Arctic sea ice domain (70°-83°N, 30°E-180°). This figure was created by the NCAR Command Language (NCL) Version 6.6.2 (https://www.ncl.ucar.edu/). Source data are provided as a Source Data file.

experience considerably higher-than-normal sea ice during autumn.

## Physical process for the linkage of summer APO to autumn sea ice in the eastern Arctic

To address the physical link, we investigate the summer APO-related dynamic and thermodynamic processes. Figure 2a, b show 200 hPa and 850 hPa horizontal wind anomalies in autumn regressed against the summer APO index, respectively. Associated with the positive APO phase, an equivalent barotropic structure characterized by strong anticyclonic circulation anomalies dominating the Barents-Kara-Laptev Seas is observed. In contrast, anomalous anticyclonic and cyclonic circulations reside in the upper and lower troposphere of the East Siberian Sea, respectively. The anticyclonic circulation anomaly over the Barents-Kara-Laptev Seas and the cyclonic circulation

anomaly over the East Siberian Sea in the lower troposphere both favor the advection of warm and moisture airflows into the eastern Arctic, leading to significant increases of humidity in situ (Fig. 2c). As an important greenhouse gas, the increase in the lower-tropospheric moisture can further enhance downwelling longwave radiation (DLR, Fig. 2d) and warm up surface air temperature (SAT, Fig. 2e)[56–59], which are key to the eastern Arctic sea ice variability. Correlations indicate that the low-level humidity correlates well with the DLR ($r = 0.95$, $p < 0.01$) and SAT ($r = 0.91$, $p < 0.01$) over the eastern Arctic, and the eastern Arctic-averaged low-level humidity, DLR and SAT are all highly correlated to the SICI ($r = 0.66, 0.70, 0.74$, respectively, $p < 0.01$), manifesting as increased low-level humidity, DLR and SAT conducive to autumn sea ice reduction in the eastern Arctic (Supplementary Fig. 4). The correlations of summer APO with the autumn low-level humidity, DLR, and SAT averaged over the eastern Arctic are 0.52, 0.58,

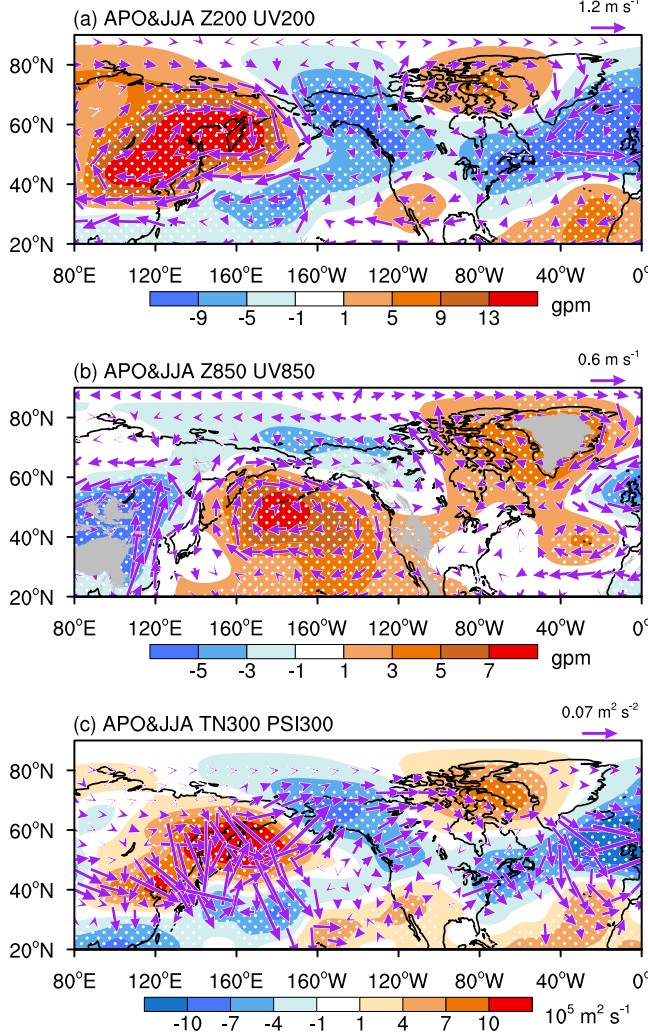

**Fig. 3 | Atmospheric pattern associated with the Asian-Pacific Oscillation (APO) in summer (June-July-August, JJA). a** 200 hPa horizontal wind (UV200, vectors, m s⁻¹) and geopotential height (Z200, shading, gpm) and (**b**) 850 hPa horizontal wind (UV850, vectors, m s⁻¹) and geopotential height (Z850, shading, gpm) in summer regressed onto the normalized summer APO index over the period 1950–2019. **c** Horizontal wave activity flux (TN300, vectors, m² s⁻²) and stream function (PSI300, shading, 10⁵ m² s⁻¹) at 300 hPa in association with the normalized summer APO index from 1950 to 2019. Areas significant above the 90% confidence level are dotted. This figure was created by the NCAR Command Language (NCL) Version 6.6.2 (https://www.ncl.ucar.edu/). Source data are provided as a Source Data file.

and 0.56 ($p < 0.01$), respectively. This suggests that the thermodynamic conditions associated with the positive summer APO are conducive to a decline in eastern Arctic sea ice in autumn.

Certainly, due to short memory of the atmosphere, the APO signal cannot persist from summer to autumn by itself. How does summer APO exert impacts on the following autumn thermodynamic processes over the eastern Arctic? We hypothesize that the North Atlantic and the North Pacific SSTs may play roles.

According to the static equilibrium relationship, warming (cooling) of air column tends to induce an increase (a decrease) in the upper-level pressure and a decrease (an increase) in the lower-level pressure via the expansion (contraction) of the air column. Thus, during the positive APO phase, the warming in Asia yields an anticyclonic circulation anomaly in the upper troposphere (Fig. 3a) and a cyclonic circulation anomaly in the lower troposphere (Fig. 3b). The situation is reversed over the North Pacific due to cooling, showing an anomalous cyclonic circulation in the upper troposphere and an

anomalous anticyclonic circulation in the lower troposphere. Furthermore, the anomalous pattern over the Asian-North Pacific sector is teleconnected with an anomalous cyclonic circulation between 40° and 60°N and an anomalous anticyclonic circulation south of it in the upper troposphere of the North Atlantic (Fig. 3a). The westerly anomalies in between indicates an intensification of the westerly jet stream over the North Atlantic, which can induce anomalous descending motion on the right-hand side of the exit of jet stream through a secondary circulation due to the effect of geostrophic deviation[60]. To compensate the downward outflow, a mass divergence is introduced, yielding an anticyclonic circulation anomaly in the lower troposphere of the mid-latitude North Atlantic (Fig. 3b).

We hypothesize the anomalous patterns in the upper troposphere of the Asian-North Pacific sector and the North Atlantic are linked via a zonal wave train propagating from the Asian-North Pacific region to the North Atlantic via North America, with positive stream function anomalies over East Asia and eastern North America and negative anomalies over the western coast of North America and the North Atlantic north of 40°N (Fig. 3c). Note that the wave activity over eastern North America is relatively weak. Such a downstream propagation of wave train was previously described in Ref. 52, but the wave activity seems weaker in the North Atlantic compared to Fig. 3c, which may be due to that it used different domains of Asia (10°–40°N, 30°–140°E) and the North Pacific (10°–40°N, 180°–90°W) to measure the APO during May–August. Ref. 61 argued that the Asian land heating induces a westward propagation of tropospheric temperatures to the Atlantic, leading to increased tropospheric temperatures over the North America–Atlantic region and an anomalous high in the troposphere of the Atlantic. It suggests a consistently varying feature of tropospheric temperatures between Asia and the North America-Atlantic sector, which differs from the APO-related tropospheric temperatures that show positive values over the extratropics of Eurasia and negative values over the extratropics of the North Pacific, North America, and the North Atlantic[48]. Thus, the above two processes may link to different phenomena although both affect atmospheric circulations over the North Atlantic. However, the relative importance of the upstream and downstream wave propagations on the North Atlantic climate remains an open issue, which needs more studies in the future.

The low-level anticyclonic anomalies over the subtropical North Pacific and North Atlantic hint at enhanced subtropical highs, since these regions are climatologically controlled by high pressure system. Underlying the intensification of subtropical highs, warm SST anomalies can be induced[51,61–64] (Fig. 4a). Because of long memory of the SSTs, these warming anomalies can persist from summer to autumn (Fig. 4b). We define the NP_sst and NA_sst indices, respectively, as the area-averaged SSTs over the North Pacific (30°–50°N, 145°E–150°W) and the North Atlantic (30°–50°N, 75°–25°W). The lead-lag correlations show that the correlations of summer APO leading the NA_sst (NP_sst) are higher than that of summer APO lagging the NA_sst (NP_sst) (Fig. 4c, d). This finding suggests that the main physical process of air-sea interaction in the two regions is the APO forcing on the ocean rather than its response to the oceanic forcing. This is consistent with the result from the modeling studies[61,62], which showed that when a NA_sst-like or NP_sst-like anomaly is imposed in a general circulation model, a local circulation response is modeled and no APO-like pressure anomalies are generated in the Asia-Pacific sector.

Previous studies documented the role of the North Atlantic and the Pacific SSTs in the Arctic sea ice change[46,65–67]. Thus, the APO resultant SST warming in autumn is expected to affect the eastern Arctic sea ice through influencing atmospheric circulations. By examining the autumn NA_sst-related wave activity flux (Fig. 5a), we notice a well-organized atmospheric wave train propagating from the North Atlantic northeastward to the Barents-Kara-Laptev Seas and then shifting southeastward to central Eurasia in the upper troposphere.

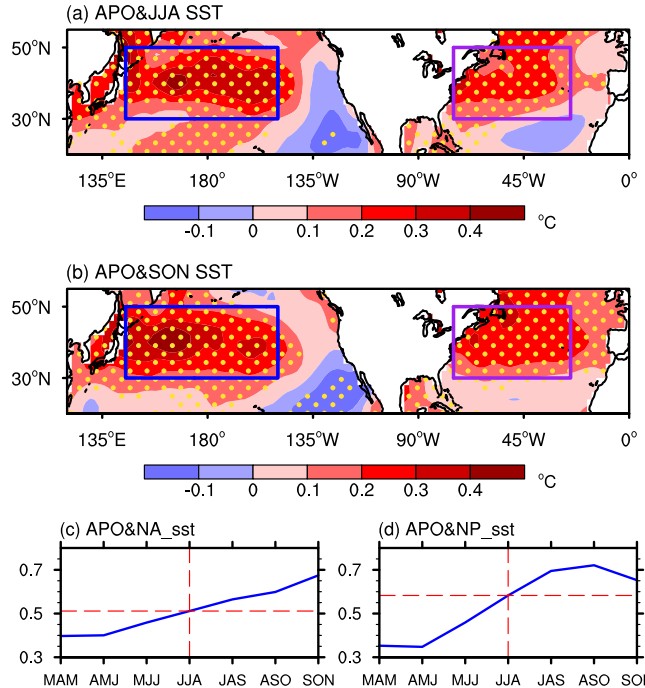

**Fig. 4 | Evolution of Asian-Pacific Oscillation (APO)-associated sea surface temperature (SST) changes from summer to autumn. a** Summer and **b** autumn SSTs (°C) regressed onto the normalized summer APO index over the period 1950–2019. Areas significant above the 90% confidence level are dotted. The blue and purple boxes in (**a**) and (**b**) outline the North Pacific (30°–50°N, 145°E–150°W) and North Atlantic (30°–50°N, 75°–25°W) domains used for the definition of NP_sst and NA_sst, respectively. **c–d** Lead-lag correlations of summer APO with **c** NA_sst and **d** NP_sst. The dashed vertical lines in **c** (**d**) represents the simultaneous correlation of APO with the NA_sst (NP_sst) in JJA, to the left of which indicates the NA_sst (NP_sst) leading the APO and to the right of which indicates the APO leading the NA_sst (NP_sst). MAM, AMJ, MJJ, JJA, JAS, ASO, and SON indicate March–April–May, April–May–June, May–June–July, June–July–August, July–August–September, August–September–October, and September–October–November, respectively. This figure was created by the NCAR Command Language (NCL) Version 6.6.2 (https://www.ncl.ucar.edu/). Source data are provided as a Source Data file.

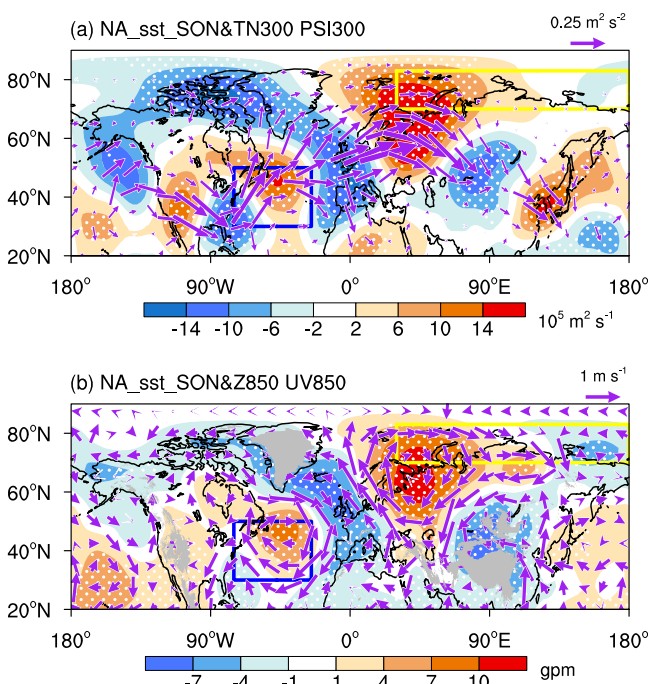

**Fig. 5 | Atmospheric pattern associated with the mid-latitude North Atlantic sea surface temperatures (SSTs) in autumn (September-October-November, SON). a** Horizontal wave activity flux (TN300, vectors, m² s⁻²) and stream function (PSI300, shading, 10⁵ m² s⁻¹) at 300 hPa in association with the normalized NA_sst index (the area-averaged SSTs over the North Atlantic (30°-50°N, 75°-25°W)). **b** 850 hPa horizontal winds (UV850, m s⁻¹) and geopotential height (Z850, shading, gpm) regressed onto the normalized NA_sst index. Areas significant above the 90% confidence level are dotted. The blue and yellow boxes outline the North Atlantic (30°–50°N, 75°–25°W) and the eastern Arctic sea ice (70°–83°N, 30°E–180°) domains, respectively. This figure was created by the NCAR Command Language (NCL) Version 6.6.2 (https://www.ncl.ucar.edu/). Source data are provided as a Source Data file.

Accordingly, the Barents-Kara-Laptev Seas are occupied by an anomalous anticyclonic circulation. The strong anticyclonic circulation anomaly extends from the upper troposphere to the lower troposphere, displaying a barotropic structure (Fig. 5). In addition, a relatively weak cyclonic circulation anomaly appears over the lower troposphere of the East Siberian Sea (Fig. 5b). Generally, the NA_sst-related pattern shown in Fig. 5b highly resembles that associated with the positive phase of summer APO (Fig. 2b), favoring the intrusion of warm and moisture airflows into the eastern Arctic. As a consequence, the low-level moisture, DLR and SAT increase in the eastern Arctic, leading to a decrease of sea ice (Supplementary Fig. 5). The simultaneous correlations of the NA_sst with the low-level moisture, DLR, and SAT averaged over the eastern Arctic and the SICI in autumn are 0.49, 0.50, 0.45, and 0.47 ($p < 0.01$), respectively.

In comparison, autumn NP_sst is related to a Pacific-North America (PNA)-like wave train (Supplementary Fig. 6a), which emanates from the mid-latitude North Pacific towards North America and shows less conspicous effects on atmospheric circulations over the eastern Arctic. Instead, anomalous cyclonic circulation over the East Siberian Sea which accompanies the anticyclonic circulation anomaly over the North Pacific in the lower troposphere (Supplementary Fig. 6b) may favor the transportation of warm and moisture airflows from lower latitudes to the eastern Arctic. The simultaneous correlations of the

NP_sst with the eastern Arctic-averaged low-level moisture, DLR, and SAT in autumn are 0.39, 0.39, and 0.33, respectively, all of which are lower than the counterparts ($r = 0.49$, 0.50, 0.45, respectively, $p < 0.01$.) with autumn NA_sst.

Considering the collinearity ($r = 0.65$, $p < 0.01$) between autumn NP_sst and NA_sst time series which implies that the SSTs over the two domains are dependent on each other, partial correlations are performed to identify unique and independent influence. When the effect of the NA_sst signal is controlled, the partial correlations between autumn NP_sst and the eastern Arctic low-level moisture ($r = 0.11$), DLR ($r = 0.09$), and SAT ($r = 0.05$) become insignificant. However, in the absence of the NP_sst signal, the relationships of autumn NA_sst with the eastern Arctic low-level moisture ($r = 0.33$), DLR ($r = 0.35$), and SAT ($r = 0.33$) remain substantial. This result suggests a primary role of the North Atlantic SSTs in linking summer APO to autumn thermodynamic processes and hence the sea ice variability in the eastern Arctic.

## Discussion

The potential contribution of climate system internal variability to the Arctic sea ice loss is a highly concerning issue. Recent studies have disclosed some climate modes of internal variability that act as drivers for the variability of Arctic sea ice. Our work advances this understanding by highlighting a physical driver, i.e., APO, an atmospheric mode of internal variability characterized by out-of-phase variations in UTT between Asia and the North Pacific. The North Atlantic SSTs are proposed to play a role in bridging such a lagged relationship. A positive APO phase in summer can give rise to warmer SSTs in the mid-

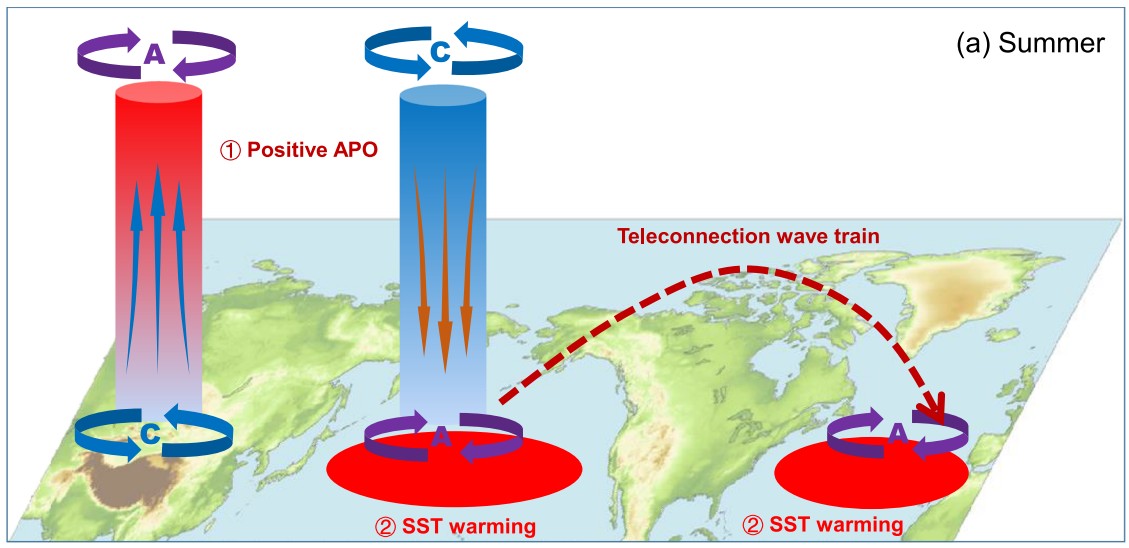

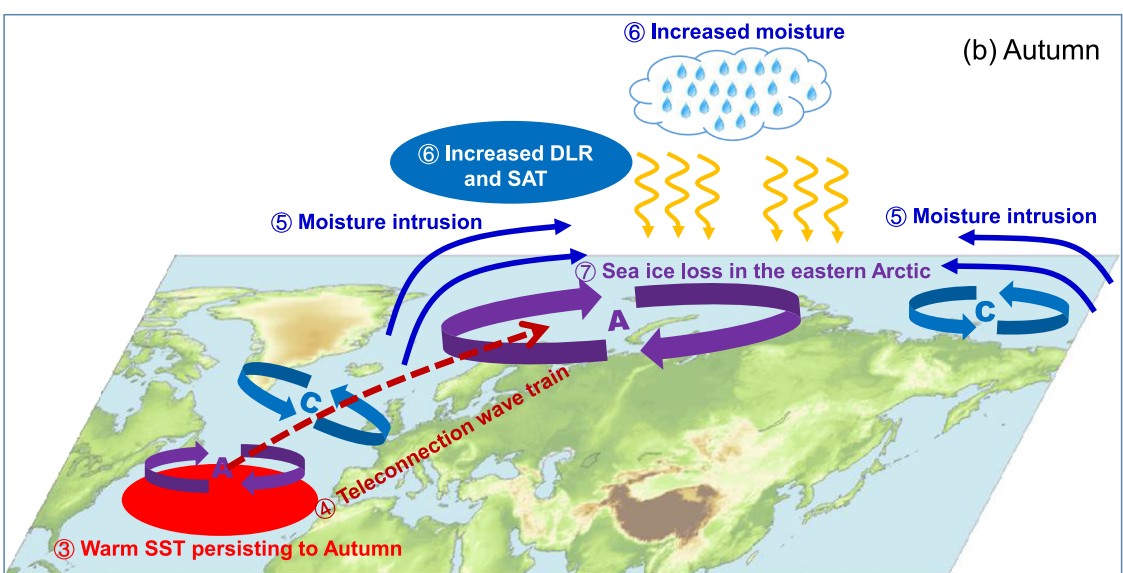

**Fig. 6 | The process of summer Asian-Pacific Oscillation (APO) influencing the following autumn sea ice in the eastern Arctic. a** Changes in summer atmospheric circulations and sea surface temperatures (SSTs) related to the positive phase of summer APO. **b** Changes in atmospheric circulations and thermodynamic processes associated with the mid-latitude North Atlantic SSTs during autumn. A and C indicate anticyclonic and cyclonic circulations, respectively.

latitude North Atlantic. Such a warming persists to autumn and then influences the eastern Arctic thermodynamic processes (i.e., increased lower-tropospheric moisture, DLR and SAT) via a teleconnection wave train emanating from the North Atlantic to the eastern Arctic, consequently resulting in sea ice loss in the eastern Arctic. The physical process for the impact of summer APO on autumn sea ice in the eastern Arctic is summarized in Fig. 6.

To complement the observational results and verify the proposed physical process, we perform analysis using the CESM-LE experiments, in which the members that can reproduce a significant relationship between summer APO and autumn SIC in the eastern Arctic are selected as an ensemble named BMME (see "Data and Methods" section). Overall, the BMME simulated patterns of the first leading MCA mode (accounting for 50.1% of the squared covariance) between summer UTT in the Asian-Pacific sector and autumn SIC in the eastern Arctic, with the temporal expansion series correlated at $r = 0.58$ ($p < 0.01$) (Supplementary Fig. 7), are broadly comparable to the observation (Fig. 1). The physical processes derived from the observation are also visible in the BMME simulation, which shows that associated with the positive APO pattern is the warming of summer SSTs in the mid-latitudes of the North Pacific and North Atlantic and the SST warming can persist to the subsequent autumn (Supplementary Fig. 8). The North Atlantic SST warming in autumn further excites a wave train and triggers barotropical anticyclonic anomalies over the Barents-Kara-Laptev Seas (Supplementary Fig. 9), beneficial for the increase in the low-level moisture ($r = 0.34$, $p < 0.05$) and SAT ($r = 0.27$, $p < 0.05$) and hence reducing sea ice in the eastern Arctic. In comparison, for the WMME simulation which simulates a significant positive correlations between summer APO and autumn SIC in the eastern Arctic that are contrary to the observation (see "Data and Methods" section), the persistent SST warming in the North Atlantic is substantially reduced and no longer statistically significant ($p < 0.05$) in autumn (Supplementary Fig. 10). This discrepancy in the results between the BMME simulation and the WMME simulation suggests the

importance of the North Atlantic SSTs in the lagged impact of summer APO on autumn SIC in the eastern Arctic. However, the reasons for the differences between the BMME and WMME in simulating the relationship of the APO with the North Atlantic SST are unclear, which deserves further in-depth investigation. One candidate reason may be due to different seasonal evolution of the oceanic temperatures associated with the APO. Previous studies showed that the oceanic temperature anomalies extending to deep mixed layer are essential to the seasonal persistence of SST anomalies via advection and mixing processes[68–70]. In the BMME simulation (Supplementary Fig. 11a), the APO-related oceanic warming in the North Atlantic extends from the sea surface to a depth of 150 meters in summer. This deep sub-surface warming lasts from summer to autumn, conducive to the maintenance of SST warming. In contrast, for the WMME simulation (Supplementary Fig. 11b), a shallow oceanic warming which only extends to a 50-meter depth is modeled in summer, failing to yield SST warming in autumn.

These findings have implications for present and future changes of Arctic sea ice in the context of global warming. Given strong correlations and process-based physical linkage between summer APO and the subsequent autumn thermodynamic processes and sea ice variability over the eastern Arctic, a recent decadal shift of the summer APO from the negative phase to the positive phase (Fig. 1c, Supplementary Fig. 2b) tends to accelerate autumn sea ice loss caused by global warming in the eastern Arctic. Under future warming scenarios, the autumn Arctic sea ice loss is projected to continue due to the effect of greenhouse gases[71], with more rapid changes for higher warming levels[1,72]. This declining rate may be accelerated or counteracted by future changes in climate system internal variability. The CMIP5 and CMIP6 project a weakening in summer APO toward the end of this century in a warmer world[73,74]. Given the results presented in this study, such a tendency is speculated to increase autumn sea ice in the eastern Arctic, which may, to some degree, counteract the sea ice loss resulting from the anthropogenic warming and thus is a source of uncertainty in projections of Arctic sea ice, although its relative importance is unclear. From this perspective, better representation of the APO in global climate models may reduce the uncertainty in the projected Arctic sea ice change. It is also worth noting that our proposed candidate mechanism underlying the linkage between summer APO and autumn sea ice in the eastern Arctic focuses on the pathway via the North Atlantic SSTs. Given that the North Atlantic SST driven SIC variability shows lower correlation scores than the APO to SIC, this reminds us that other processes may also act in their linkage, which deserves further in-depth investigation in the future in order to get a broad picture.

## Methods

### Reanalysis data
The monthly SIC with a resolution of 1.0°×1.0° is obtained from the Met Office Hadley Centre Sea Ice and SST dataset (HadISST1)[75]. The SIC was estimated from satellite retrievals after 1978, prior to which it was reconstructed from different sources of data. The monthly SST dataset used is the Extended Reconstructed SST version 5 (ERSST v5) at 2.0° × 2.0° from the National Oceanic and Atmospheric Administration (NOAA)[76]. The atmospheric reanalysis data at 2.5°×2.5° are provided by the National Centers for Environmental Prediction/National Center for Atmospheric Research (NCEP/NCAR)[77].

### Community earth system model large ensemble (CESM-LE) simulations
Large "initial condition" ensemble experiments have been developed and widely used to understand the impact of internal variability in recent years. To validate the observational analysis, we use the CESM-LE simulations[78], in which 40 members performed by the Community Earth System Model (CESM) with atmosphere-ocean-land-sea ice fully coupled at 0.9°×1.25° (latitude×longitude). Each member is subject to

the same historical forcing from 1920 to 2005, but with a small random noise perturbation to their initial air temperature fields. In this study, three members that reproduce a significant negative correlation ($p < 0.1$ with the Monte Carlo test[79]) between summer APO and autumn SIC averaged over the eastern Arctic for the period 1950–2005, which is consistent with the observation, are selected as an ensemble named BMME, given that this relationship is not captured by each of the 40 members. For comparison, four members that show a significant positive correlation ($p < 0.1$ with the Monte Carlo test[79]) between summer APO and autumn SIC over the eastern Arctic for the period 1950–2005, which is contrary to the observation, are selected as an ensemble named WMME. The ensemble mean is firstly calculated by individual members and then averaged with equal weight.

In addition to the time series of the leading MCA of UTT (red line in Fig. 1c), another convenient method to construct the APO index is referring to the locations of the teleconnection centers[48], because the calculation by this method is independent on the length of the study time. According to Supplementary Fig. 2a, if we define the arithmetic difference of the UTT between the Asian region (30°–55°N, 80°–135°E) and the North Pacific region (30°-55°N, 170°E-125°W) as the APO index, this index is highly correlated with both the PC1 series of the UTT pattern ($r = 0.95$, $p < 0.01$) and the time series of the leading MCA of UTT ($r = 0.95$, $p < 0.01$) over the Asian-Pacific sector. To facilitate analysis, this definition is adopted to measure the APO variability in the simulations.

### Statistical analyses
The MCA method[80], performed by singular value decomposition of the covariance matrix between two fields, is adopted to identify dominant modes of co-variability between summer UTT over the Asian-Pacific region and autumn SIC in the eastern Arctic. The leading pair of singular vectors and the associated expansion series obtained by projecting the singular vectors onto the original fields indicate the spatial pattern and temporal variation of the two fields that are optimally coupled. The squared singular value explains the contribution of the pair of singular vectors to the total covariance between them.

The horizontal wave activity flux (**W**) is applied to detect the energy propagation of quasi-stationary Rossby waves, which is calculated in terms of the equation[81]:

$$\mathbf{W} = \frac{p}{2|U|} \begin{bmatrix} U\left(\psi_x'^2 - \psi'\psi_{xx}'\right) + V(\psi_x'\psi_y' - \psi'\psi_{xy}') \\ U\left(\psi_x'\psi_y' - \psi'\psi_{xy}'\right) + V(\psi_y'^2 - \psi'\psi_{yy}') \end{bmatrix} \quad (1)$$

in which $\psi'$ is the perturbation stream function, $|U|$ is the horizontal wind speed, $U$ ($V$) is the zonal (meridional) component of the basic flow for the whole period, and $p$ is the pressure divided by 1000 hPa. The anomalous perturbation stream function associated with the given index is calculated by the regression.

Regression, correlation, EOF, and wavelet transform coherence (WTC) analyses are also employed. The statistical significance of the results for regression, correlation, and WTC analyses is determined by the Monte Carlo test[79] with 1000 simulations. In the Monte Carlo test, the null hypothesis is that two variables X and Y are not related (i.e., the correlation coefficient $r = 0$), and the alternative hypothesis assumes a significant relationship between the two variables. For a given significance level $\alpha = 0.1$, for example, we first randomly sample X 1000 times, and then calculate $r$ values of each sample with Y. The obtained 1000 simulated $r$ values are ranked from the smallest to the largest, and the 5th (95th) percentile value is used as the critical value of $r_{0.05}$ ($r_{0.95}$). If the actual $r$ is smaller than $r_{0.05}$ or greater than $r_{0.95}$, the null hypothesis is rejected and we consider it significant at $\alpha = 0.1$. The time period used for analysis in this study is from 1950 to 2019. Before analysis, all the data are linearly detrended.

## Data availability

The HadISST1 SIC and the NOAA ERSST v5 data are obtained from https://www.metoffice.gov.uk/hadobs/hadisst/data/download.html and https://psl.noaa.gov/data/gridded/data.noaa.ersst.v5.html, respectively. The NCEP/NCAR reanalysis data are downloaded from https://psl.noaa.gov/data/gridded/data.ncep.reanalysis.derived.html. The CESM-LE simulations are accessible at https://www.cesm.ucar.edu/community-projects/lens/data-sets. The data generated for all figures of this study are provided in the Source Data file. Source data are provided with this paper.

## Code availability

All figures are created by the NCAR Command Language (NCL) Version 6.6.2 (available at https://www.ncl.ucar.edu/), or the MATLAB Version R2021B (available at https://ww2.mathworks.cn/products/matlab.html). The code for MCA analysis is available at https://www.ncl.ucar.edu/Document/Functions/Built-in/svdstd.shtml. The WTC analysis is performed by the MATLAB wavelet coherence toolbox (available at https://www.glaciology.net/publication/2004-12-24-application-of-the-cross-wavelet-transform-and-wavelet-coherence-to-geophysical-time-series/). Other computer codes that support the analysis within this paper are available from the corresponding author on request.

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

## Acknowledgements

This research was jointly supported by the National Natural Science Foundation of China (42025502 to B.Z.) and the Postgraduate Research

& Practice Innovation Program of Jiangsu Province (KYCX22_1140 to Z.S.).

## Author contributions

B.Z. conceived this study. Z.S. performed data analysis and plotted figures under B.Z.'s instruction. Z.Y., X.X., B.S., P.H., and H.C. contributed to the discussion of the mechanisms. B.Z. wrote the manuscript with the contribution from Z.S., Z.Y., X.X., B.S., P.H., and H.C.

## Competing interests

The authors declare no competing interests.

## Additional information

**Peer review information** : *Nature Communications* thanks Michel Tsamados and the other, anonymous, reviewer(s) for their contribution to the peer review of this work. A peer review file is available.

