## [Peer Review File · Nature Communications]

Recent autumn sea ice loss in the eastern Arctic enhanced by summer Asian-Pacific OscillationREVIEWER COMMENTS

Reviewer #1 (Remarks to the Author):

Review of study of Autumn sea ice loss in the eastern Arctic mediated by summer Asian-Pacific Oscillation by Zhou et al

The authors provide a clear analysis of a new (?) Asian-Pacific Oscillation driver of Eastern sea ice variability on annual to decadal timescales. The arguments are well structured in a series of clear figures that 1) identify teleconnections between APO and SIC 2) propose mechanistic linkages between APO, SST and SIC 3) contextualise the results against a CMIP class model (CESM). I was not aware of the role of APO in controlling SIC and as such find the paper interesting. I would consider the paper for publication but there are several major issues that I would like the authors to consider and address beforehand (see below). Overall I am worried that the importance of the findings are somewhat exaggerated in that they do not give sufficient credit to similar results in the literature (even if APO was not mentioned explicitly there), they don't provide an assessment of the relative importance of this APO-SIC teleconnection against other well known drivers of sea ice variability, they confuse variability on long term changes and inter-annual variability in their analysis and use a dataset that is not observation per se.

General Comments:

- The authors should redo their analysis on the satellite record period directly on the satellite data of sea ice concentration / extent. Or at least characterise the realism of the analysis product they use. It is not clear as well which version of HadISST they use. Quote also <https://agupubs.onlinelibrary.wiley.com/doi/full/10.1002/2013JD020316> and product link.
- The physical processes leading to correlation between summer APO and Autumn SIC are hinting at a pathway via SST but the authors do not demonstrate that causality directly nor do they reference the literature on SST to SIC teleconnections that is very rich in both Pacific and Atlantic sectors.
- I find the literature review very partial with a lot of references ignored that have looked at very similar SST to SIC teleconnections, for example a few below but there are many more. How does this work relate to this literature and what is new (apart from the APO definition and use).

Bonan et al (2020) <https://doi.org/10.1029/2019GL085666>

Liu et al (2021) <https://www.nature.com/articles/s41467-021-21830-z>

Kapsch et al (2013) <https://www.nature.com/articles/nclimate1884>

- Can you expand as to why the SST driven SIC variability is showing lower correlation scores than the APO to SIC.
- One more proof reading required to iron out some typos and small grammatical errors.
- The procedure of selection of CESM ensemble members needs to be clarified.
- Regression is done at multiannual time scales (i.e. there is no detrending) and yet mechanisms are discussed at inter-annual scale. Are you claiming that APO drives inter annual variability or variability on longer time scales? Clarify
- Your extrapolations to a negative feedback on sea ice decline by the end of the century seem a little weak. In particular against the context of rapid Arctic amplification and disintegration of the sea ice. In general something that is missing from your analysis is the relative importance of your findings to other more well known drivers of sea ice variability (i.e. AO) and decline (CO2 emissions). See for examples papers by Dirk Notz and Julienne Stroeve.
- It is well known that regression between indices (AO, etc...) and SIC are not stationary and change drastically depending on the duration of the time period studied. The authors should assess their results over the satellite period only and use satellite products only. I would also like to see more analysis of the time scale over which their teleconnections are effective. You could for example produce a spectral analysis and look at drivers of the time series for different amounts of smoothing.

Would the authors care to expand to teleconnections between other seasons and if not explain why summer to autumn is the most clear result.

Specific Comments:

- Fig 2f is quoted before Fig2 a-e
- Fig 2a and 2b hard to see the land contours
- Fig 2 needs clarifying as it is not clear in the title that we are looking at regressions
- L140 define the static equilibrium relationship

Reviewer #2 (Remarks to the Author):

See attached.

Reviewer #3 (Remarks to the Author):

Review of the paper

'Autumn sea ice loss in the eastern Arctic mediated by summer Asian-Pacific Oscillation'

by B. Zhou, Z. Song, Z. Yin, X. Xu, B. Sun, P. Hsu, and H. Chen
submitted to 'Nature Communication'

The aim of the manuscript is to analyse the role of a specific atmospheric variability pattern in summer, namely the Asian-Pacific Oscillation (APO), as a potential driver of autumn sea ice variability in the eastern Arctic. The authors proposed a chain of dynamical processes linking variations in the APO to autumn sea ice loss in the eastern Arctic, whereby the recent trend of the summer APO towards its positive phase may have contributed to accelerated autumn sea ice loss in the eastern Arctic in autumn.

This study continues a series of papers by the lead author dealing with the APO and its relationship with climate variability over the North Pacific or North Atlantic, respectively.

This study contributes to the large body of literature investigating the connections between the Arctic and the mid-latitudes. While the majority of papers investigate how the dramatic Arctic changes in autumn and winter, in particular sea ice retreat and increases in near-surface temperature, may trigger tropospheric and stratospheric pathways from the Arctic to the mid-latitudes, this study contributes to the less studied link between changes in the mid-latitude circulation and the Arctic climate system, in particular Arctic sea ice retreat in autumn (which in turn may trigger the above-mentioned Arctic-mid-latitude linkages in autumn and winter). This study is therefore timely and relevant.

The manuscript is clear and mostly easy to follow, but in my opinion several aspects of the proposed chain of dynamical processes linking variations in the APO to autumn sea ice loss in the eastern Arctic lack evidence, in particular the link from the anomalous circulation over the Asian-North Pacific sector to the anomalous circulation in the mid-latitudes of the North Atlantic. Because this is a key link in connecting SUMMER APO to AUTUMN sea-ice loss, in my view it is necessary to provide more evidence in this regard.

In addition, the description of the methods needs to be improved to ensure that the reader can understand all the analyses and hence the subsequent conclusions. This explicitly includes the analysis of the modelling results, in particular the choice of the members for the sub-ensemble.

At this stage, the submitted manuscript needs careful and major revision.

Major comments:

(1) My main concern is the missing evidence for the link from the anomalous circulation over the Asian-North Pacific sector to the anomalous circulation in the mid-latitudes of the North Atlantic. This is critical, because this is a key link in connecting SUMMER APO to AUTUMN sea-ice loss. The authors argue that " the anomalous pattern over the Asian-North Pacific sector is teleconnected with an anomalous anticyclonic circulation in the mid-latitudes of the North Atlantic through a zonal wave train" (L145-147) with reference to a previous paper by the lead author (Zhou and Wang, JGR, 2015). In my opinion, Fig. 3 alone does not show the real extent of the suggested wave train. It is not clear, which centers of action are really connected via wave activity fluxes, nor the start and end of the wave train. In the referenced earlier study (Fig. 9a in Zhou and Wang, JGR, 2015), wave activity fluxes for the suggested wave train are shown, but in my view, the wave train ends over eastern North America due to only very weak wave activity fluxes originating from the stream function anomaly over eastern North America towards the mid-latitude North Atlantic. I would like to ask the authors to provide stronger evidence for the suggested link from the anomalous circulation over the Asian-North Pacific sector to the anomalous circulation in the mid-latitudes of the North Atlantic.

Moreover, in L160, the authors give the reference Zhao et al., JC (2012). These authors argue that "...the westward propagation of the positive anomalies results in an increase of tropospheric temperature over the North America–Atlantic region. As seen later from section 4d, this westward propagation may be forced by Asian surface heating anomalies. Accordingly, significant positive anomalies appear in the troposphere over North America and the Atlantic, Thus the tropospheric temperature over North America and the Atlantic is high (low) when Asian tropospheric temperature is high (low), suggesting a consistently varying feature of tropospheric temperature between Asia and the North America–Atlantic sector." (Zhao et al., JC (2012), p. 6600). Here, a different mechanisms is suggested. Could the authors should analyse and discuss these different mechanisms.

(2) Parts of the "Data and Methods" section have to be improved by providing necessary information to enable the reader to better understand the analyses performed in the study.

With regard to the simulations: more details have to be given, how those ensemble members have been selected, which simulate an summer APO- autumn SIC relation, how many ensemble members from the 40 members simulate the APO-SIC relation, and how the ensemble patterns shown in supplementary Figs. 6 and 7 have been calculated. What are potential reasons that not all ensemble members reproduce the observed summer APO- autumn SIC relation?

With regard to statistical analyses: Please provide information, how the perturbations (needed for the calculation of the wave activity fluxes) have been calculated, since the definition of the basic flow has an impact on the calculated wave fluxes.

(3) The authors have used seasonally (3 months) averaged data for their analyses. How robust are their results when using data with higher temporal resolution (monthly means or means over pentads)?

Minor comments

(1) General comment for figures 2, 3, 5, S4, S7: I strongly recommend to show the same area in all these figures for better comparability, and in particular to extend the area shown in fig. 5 to the west (to allow a better identification of the begin and end of wave trains).

(2) Introduction L32-67: For some statements, newer references should be added, I here give one example: Arctic amplification: Arctic warming nearly four times faster than the globe (Rantanen et al., 2022). Please check newer publications also for other statements.

Rantanen, M., Karpechko, A. Y., Lipponen, A., Nordling, K., Hyvärinen, O., Ruosteenoja, K., ... & Laaksonen, A. (2022). The Arctic has warmed nearly four times faster than the globe since 1979. *Communications Earth & Environment*, 3(1), 168.

(3) L39: What means etc.?

(4) L34-L44: Please give also refernces for the uncertainties and knowledge gaps with regard to the impacts of Arctic changes on the local and remote weather and climate.

(5) L65: "the reanalysis and simulation data" please, be more specific.

(6) L70 and following: I strongly recommend to reformulate the frist sentences of the results section to allow the reader an easier entry into the results section.

(7) L74-L76: Please give lat-lon information for the Asian-Pacific region in the text.

(8) L74-L76: How sensitive are the results of MCA (or EOF) on the chosen area for the Asian-Pacific region? Does the dipole pattern of the APO appear as first mode for a larger region as well?

(9) L98: How are SIC anomalies calculated?

(10) L488/Fig.1: Please indicate the correlation coefficient between the time series in Fig. 1c. Same for Fig. S5c.

(11) Fig. S1: The shown timeseries for the APO (PC1 of EOF1 mode over Asian-Pacific sector) deviates from that shown in Zhou et al., 2015 over the period 1965-2000. In Zhou et al. (2015), Fig. 3, I do see interannual variability of the index with no preference of a negative phase, whereas the timeseries shown in Fig. S1 does show a prolonged negative phase from ~1965 to ~2000. Please, could you comment on these differences?

Response to the Reviewer #1

The authors provide a clear analysis of a new (?) Asian-Pacific Oscillation driver of Eastern sea ice variability on annual to decadal timescales. The arguments are well structured in a series of clear figures that 1) identify teleconnections between APO and SIC 2) propose mechanistic linkages between APO, SST and SIC 3) contextualise the results against a CMIP class model (CESM). I was not aware of the role of APO in controlling SIC and as such find the paper interesting. I would consider the paper for publication but there are several major issues that I would like the authors to consider and address beforehand (see below). Overall I am worried that the importance of the findings are somewhat exaggerated in that they do not give sufficient credit to similar results in the literature (even if APO was not mentioned explicitly there), they don't provide an assessment of the relative importance of this APO-SIC teleconnection against other well known drivers of sea ice variability, they confuse variability on long term changes and inter-annual variability in their analysis and use a dataset that is not observation per se.

Reply: Positive comments and helpful suggestion from the reviewer are highly appreciated. We have carefully considered all the comments and made substantial revisions. The revisions in the manuscript are highlighted by red color.

General Comments:

- The authors should redo their analysis on the satellite record period directly on the satellite data of sea ice concentration /extent. Or at least characterise the realism of the analysis product they use. It is not clear as well which version of HadISST they use. Quote also <https://agupubs.onlinelibrary.wiley.com/doi/full/10.1002/2013JD020316> and product link.

Reply: The sea ice concentration (SIC) we use is from the Met Office Hadley Centre Sea Ice and SST dataset, HadISST1. The SIC was estimated from satellite retrievals after 1978, prior to which it was reconstructed from different sources of data. This has been clarified in the subsection "Reanalysis data" of the revised manuscript (*Please see Lines 280-2823*).

According to the comment, we have also redone the analysis over the satellite record

period (1980-2019), and a similar result can be generally obtained (*Please see Lines88-90 and Supplementary Fig. 1*). We also redo the analysis using the SIC data of the HadISST v2.2 (Titchner and Rayner, 2014) as mentioned above. The results shown in Fig. R1 below resemble that of Fig. 1 a-c in the manuscript.

Fig. R1: a–b Spatial patterns of the first leading MCA mode for a summer UTT over the Asian-Pacific sector and b autumn SIC in the eastern Arctic based on the HadISST.2.2 dataset over the period 1950–2019. Areas significant above the 90% confidence level are dotted. c Time series of the UTT (red line) and SIC (blue line) patterns shown in a and b, respectively.

- The physical processes leading to correlation between summer APO and Autumn SIC are hinting at a pathway via SST but the authors do not demonstrate that causality directly nor do they reference the literature on SST to SIC teleconnections that is very rich in both Pacific and Atlantic sectors.

Reply: According to the comment, we have added some references on the SST to SIC teleconnections to provide more evidence explaining the causality between summer APO and autumn SIC in the revised manuscript (*Please see L179-180*).

- I find the literature review very partial with a lot of references ignored that have looked at very similar SST to SIC teleconnections, for example a few below but there are many more. How does this work relate to this literature and what is new (apart

from the APO definition and use).

Bonan et al (2020) <https://doi.org/10.1029/2019GL085666>

Liu et al (2021) <https://www.nature.com/articles/s41467-021-21830-z>

Kapsch et al (2013) <https://www.nature.com/articles/nclimate1884>

Reply: We have added these and more references in the revised manuscript. Our work, based on these literatures, advances the understanding of Arctic sea ice loss by taking a step forward to summer signal.

- Can you expand as to why the SST driven SIC variability is showing lower correlation scores than the APO to SIC.

Reply: Our proposed candidate mechanism underlying the linkage between summer APO and autumn sea ice in the eastern Arctic focuses on the pathway via the North Atlantic SSTs. Given the complexity of climate system, other processes may also act in their linkage, which may explain that the SST driven SIC variability is showing lower correlation scores than the APO to SIC. The discussion for this issue has been added in the revised manuscript (*Please see L272-278*).

- One more proof reading required to iron out some typos and small grammatical errors.

Reply: A careful check has been carried out to fix typo and grammatical errors.

- The procedure of selection of CESM ensemble members needs to be clarified.

Reply: It has been clarified in the subsection “Community Earth System Model Large Ensemble (CESM-LE) Simulations”. (*Please see L296-303*)

- Regression is done at multiannual time scales (i.e. there is no detrending) and yet mechanisms are discussed at inter-annual scale. Are you claiming that APO drives inter annual variability or variability on longer time scales? Clarify

Reply: Note that all the data are detrended before analysis in our study (*Please see L334-335 for this statement*). Thus, there is no influence of the trends. The summer

APO drives sea ice variability on interannual-to-decadal time scales. This has been clearly stated in the revised manuscript (*Please see L114-119*).

- Your extrapolations to a negative feedback on sea ice decline by the end of the century seem a little weak. In particular against the context of rapid Arctic amplification and disintegration of the sea ice. In general something that is missing from your analysis is the relative importance of your findings to other more well known drivers of sea ice variability (i.e. AO) and decline (CO₂ emissions). See for examples papers by Dirk Notz and Julienne Stroeve.

Reply: I agree with the reviewer that the external forcing (CO₂ emissions) is the main driver for the projected Arctic sea ice loss. In the revised manuscript, we have rewritten this part to make it more balance, which is as: “Under future warming scenarios, the autumn Arctic sea ice loss is projected to continue due to the effect of greenhouse gases⁶⁷, with more rapid changes for higher warming levels^{1,68}. This declining rate may be accelerated or counteracted by future changes in climate system internal variability. The CMIP5 and CMIP6 project a weakening in summer APO toward the end of this century in a warmer world^{69,70}. Given the results presented in this study, such a tendency is speculated to increase autumn sea ice in the eastern Arctic, which may, to some degree, counteract the sea ice loss resulting from the anthropogenic warming and thus is a source of uncertainty in projections of Arctic sea ice, although its relative importance is unclear”. (*Please see L262-270*)

- It is well know that regression between indices (AO, etc...) and SIC are not stationary and change drastically depending on the duration of the time period studied. The authors should assess their results over the satellite period only and use satellite products only. I would also like to see more analysis of the time scale over which their teleconnections are effective. You could for example produce a spectral analysis and look at drivers of the time series for different amounts of smoothing. Would the authors care to expand to teleconnections between other seasons and if not explain why summer to autumn is the most clear result.

Reply: According to the comment, we have redone the analysis over the satellite record period (1980-2019). A similar result can be generally obtained. This content has been added in the revised manuscript (*Please see Lines88-90 and Supplementary*

Fig. 1). We also perform the squared wavelet coherence analysis (*Please see Supplementary Fig. 3*) in the revision to investigate the time scale over which the APO-SIC teleconnections are effective. It is indicated that they show a predominant in-phase coherency in 5-8 year, 8-12year, and 16-24 year bands (*Please see L114-117*). The reason why we care about summer APO is due to that the APO is most dominant in the summertime, which has been clarified in the revised manuscript (*Please see L61*).

Specific Comments:

- Fig 2f is quoted before Fig2 a-e

Reply: In the revised manuscript, we have moved it to Fig. 1 (as Fig. 1d).

- Fig 2a and 2b hard to see the land contours

Reply: The land contours have been thickened.

- Fig 2 needs clarifying as it is not clear in the title that we are looking at regressions

Reply: Clarified in the figure caption.

- L140 define the static equilibrium relationship

Reply: It has been defined in the revised manuscript. (*Please see L152-154*)

Response to the Reviewer #2

General comments

The authors consider the question of whether the June-July-August (JJA) Asian-Pacific Oscillation (APO) exerts an influence on the September-October-November (SON) sea-ice in the eastern Arctic. They find a significant correlation between two associated indices of 0.62, and argue that this reflects a causal forcing from the APO. The pathway is suggested to be mediated by the North Atlantic SSTs: the APO forces anomalies in North Atlantic SSTs in JJA, which persist to SON, triggering circulation anomalies that are conducive to ice loss. Decadal variability or future trends in the APO may therefore exacerbate or offset ice loss due to global warming. If true, this would be a noteworthy result, since model biases in the APO may affect their projections of sea ice loss.

I enjoyed reading this paper. It was generally easy to read and follow, and follows a clear and logical order. The figures showing wave activity fluxes are particularly neat and convincing. However, I have a few major comments which need to be addressed, in particular about the role of the North Atlantic. I am therefore recommending major revisions. I hope the authors will be able to address my comments without too much trouble.

Reply: Positive comments and helpful suggestion from the reviewer are highly appreciated. We have carefully considered all the comments and made substantial revisions. The revisions in the manuscript are highlighted by red color.

Major comments

1. The North Atlantic as a mediator

The authors argue that the correlations between the APO and Eastern Arctic sea-ice can be explained by a pathway via the North Atlantic. While physical motivation for this is given by the figures provided, some basic statistical testing is missing. The basic method for testing if something is a mediator is described e.g. on Wikipedia. The authors do not seem to state what the correlation is between their NA_SST index and Eastern Arctic sea-ice index, nor what happens if you do a multiple regression with the APO and NA_SST as predictors and eastern Arctic sea-ice as the dependent variable. Can the authors please explicitly state the NA_SST-Ice correlation and carry

out this test for mediation and discuss the results?

The correlation between APO and sea-ice is 0.62, explaining ~38% of the ice variance. While NA_SSTs and sea-ice will surely also be correlated, it can't be ruled out that the NA_SST related ice-variance is disjoint from the 38% APO-related variance. If there were actually a different pathway from APO to ice (e.g. via the stratosphere?) then this could easily be the case. The above test for mediation should help clarify this.

Reply: Thanks for useful suggestion. According to the comment, we have explicitly shown the correlation between the NA_SST and SIC indices in the revised manuscript (*Please see L194-196*). The regression of SIC with the NA_sst index is provided in Supplementary Fig. 5.

Given high linear correlations between APO and NA_SST, the multivariate linear regression may give rise to covariance problems. Instead, we use the CESM-LE ensemble simulation results to verify the proposed physical process. We select the members that can reproduce a significant negative correlation between summer APO and autumn SIC averaged in the eastern Arctic, which is consistent with the observation, as an ensemble named BMME. We also select the members that show a significant positive correlation between summer APO and autumn SIC averaged in the eastern Arctic, which is contrary to the observation, as an ensemble named WMME. Overall, the BMME simulation results are broadly comparable to the observation, showing that associated with the positive APO pattern is the warming of summer SSTs in the mid-latitudes of the North Atlantic and the SST warming can persist to the subsequent autumn (Supplementary Fig. 8). The North Atlantic SST warming in autumn further excites a wave train and triggers barotropic anticyclonic anomalies over the Barents-Kara-Laptev Seas (Supplementary Fig. 9), beneficial for the increase in the low-level moisture ($r=0.34$, $p<0.05$) and SAT ($r=0.27$, $p<0.05$) and hence reducing sea ice in the eastern Arctic. In comparison, for the WMME simulation, the persistent SST warming in the North Atlantic cannot be simulated (Supplementary Fig. 10). This discrepancy in the results between the BMME simulation and the WMME simulation suggests the importance of the North Atlantic SSTs in the lagged impact of summer APO on autumn SIC in the eastern Arctic (*Please see L233-255*). In addition, based on the partial correlation, when controlling the NA_sst signal, the APO-SICI correlation decreases from 0.62 to 0.46, which may also suggest the role of the NA_SST.

It is worth noting that our proposed candidate mechanism underlying the linkage between summer APO and autumn sea ice in the eastern Arctic only focuses on the pathway via the North Atlantic SSTs. Given that the North Atlantic SST driven SIC variability shows lower correlation scores than the APO to SIC, this reminds us that other processes may also act in their linkage, which deserves further in-depth investigation in the future in order to get a broad picture. This has been added in the revised manuscript (*Please see L272-278*)

2. Causality between APO and North Atlantic

The obvious question which arises when reading this paper is whether it's actually just the North Atlantic SSTs doing everything. The authors include an argument for why the APO is driving the NA_SSTs and not vice versa, by considering the lag-lead correlations in Fig 4c. However, I do not find this convincing. The difference depending on whether APO leads or not looks by eye to be 0.6 (APO leads) and 0.4 to 0.45 (APO lags). Because the total APO-ice correlation to be explained is only 0.62, this means that a hypothesis of the North Atlantic independently forcing some proportion of both the APO and ice variance might explain almost the correlations seen, once you allow for some noise. In addition, in the CESM_LE models, the equivalent lead-lag plot shows an even less pronounced difference between APO leading and lagging, making the authors argument less convincing still. Finally, there is considerable persistence in the SSTs, meaning that even though the forcing is on the face of it smaller, the fact that it lasts for longer also needs to be factored in.

I think this point is made more acute by the fact that the APO seems to undergo some decadal variability. By eye this variability looks consistent with that seen in the North Atlantic in the same period. This makes it more tempting to wonder if it's the North Atlantic doing the heavy lifting.

Can the authors do some further tests for whether the NA_SSTs explain what's going on? Again, you can use mediation tests (see previous comment) to examine to what extent e.g. JJA or MJJ NA_SSTs account for the APO-Ice correlations.

Reply: This issue is related to the point 1. We have carefully considered it in the revision and believe that we have addressed it to the extent we can. Below are some explanations:

First, according to the previous studies, the correlation of X leading Y higher than that

of X lagging Y suggests the influence of X on Y. Thus, the lead-lag correlations between summer APO and NA_sst may suggest the impact of summer APO on the North Atlantic SSTs. In addition, in line with the previous studies, “warm SST–ridge” relationship also implies the influence of atmosphere on ocean. From Fig. 3b and Fig. 4a in the manuscript, we can observe such a “warm SST–ridge” phenomenon. Thus, the summer APO signal may be preserved in the ocean, and then affects autumn atmospheric circulations through the feedback of the ocean. However, there is possibility that summer SST itself gives rise to the changes in autumn atmospheric circulations.

Second, the previous modeling studies indicated that the sensitive experiments with the summer SST changed cannot force the APO phenomenon, suggesting the air-sea coupled relationship is the APO forcing on the ocean.

Third, we add more analyses based on the CESM-LE simulations. We select the members that can reproduce a significant negative correlation between summer APO and autumn SIC averaged in the eastern Arctic, which is consistent with the observation, as an ensemble named BMME. In general, the BMME simulation results are broadly comparable to the observation, showing that associated with the positive APO pattern is the warming of summer SSTs in the mid-latitudes of the North Atlantic and the SST warming can persist to the subsequent autumn (Supplementary Fig. 8). For comparison, we also select the members that show a significant positive correlation between summer APO and autumn SIC averaged in the eastern Arctic, which is contrary to the observation, as an ensemble named WMME. It is interesting to find that the persistent SST warming in the North Atlantic disappears in the WMME simulation (Supplementary Fig. 10). This discrepancy in the results between the BMME simulation and the WMME simulation suggests the importance of the North Atlantic SSTs in the lagged impact of summer APO on autumn SIC in the eastern Arctic.

Certainly, the North Atlantic SSTs cannot do everything. Although our proposed candidate mechanism underlying the linkage between summer APO and autumn sea ice in the eastern Arctic focuses on the pathway via the North Atlantic SSTs, other processes may also act in their linkage given that the North Atlantic SST driven SIC variability shows lower correlation scores than the APO to SIC, which needs further in-depth investigation in the future.

All of these have been presented in the revised manuscript.

3. Statistical significance tests

The authors use a basic t-test to assess significance of correlations throughout. However, this is not appropriate when considering SSTs or sea-ice timeseries. The t-test assumes that samples are independent, but there is considerable interannual autocorrelation in SSTs and ice, meaning this assumption is not valid. In particular, the p-values reported are almost certainly much lower than they should be.

The authors should use a more appropriate method for assessing significance of correlations in cases involving SSTs and ice. The most transparent way is probably via Monte Carlo sampling using an explicit null hypothesis. For example, if correlating some atmospheric variable X and sea-ice Y, you can model X as independent Gaussian and Y as an AR-1 process. By fitting these to the data you can generate 1000 random draws and assess what correlations are expected by chance. Other methods are also possible.

Reply: According to the suggestion, we have changed the t-test to the Monte Carlo test with 1000 simulations for the significance test throughout the manuscript.

4. Decadal APO variability

There are implicit suggestions of decadal variability in the APO in the manuscript (e.g. L81 and L236). Can this point be made explicit? That is, what does prior literature suggest about decadal APO variability? Does the decadal variability seen in Figure 1c require an explanation (i.e. a source of decadal timescale external forcing) or is it consistent with random internal variability of the APO? These questions have obvious potential implications for what might happen in the future.

Reply: According to the comment, we have explicitly stated decadal variability in summer APO in the revised manuscript (*Please see L99-103*). To our knowledge, the decadal variability of APO is internal variability.

Minor comments

L1 (Title): I think the use of the word 'mediated' is not ideal, since the content of the paper will go on to discuss a pathway under which the APO's influence on sea-ice is mediated via the North Atlantic. This can cause confusion. Can the authors please

rephrase or choose another word? E.g. “enhanced” or “amplified” could work.

Reply: It has been changed to “enhanced” in the title.

L35: The extent to which the Arctic has “exerted profound impacts” on remote climate is actively debated, and there are many papers which argue it has only had a small impact. The authors should recognize the lack of consensus here. Please rephrase to clarify this, and cite some relevant papers arguing for a more minimal or unclear influence, e.g.

<https://doi.org/10.1038%2Fs41467-022-28283-y>

<https://doi.org/10.1038/s41558-020-00954-y>

<https://doi.org/10.1038/s41558-019-0662-y>

or any number of other references therein.

Reply: We have rewording this part and the lack of consensus has been clarified.

(Please see L35-36 and L44-47)

L43: “closely links to” → “is closely linked to”

Reply: Changed. *(Please see L43)*

L45: Again, the “pronounced climate impacts” are debatable. Please rephrase to reflect the lack of consensus in the literature.

Reply: We have rephrased it as “Although no consensus is reached for the climate effects of sea ice loss in the Arctic, the physical drivers for Arctic sea ice loss have sparked increasing interest”. *(Please see L48-49)*

L49: This doesn’t read very easily: “that summertime atmospheric circulation towards a stronger anticyclonic circulation”. Do you mean recent trends in summertime circulation? Or something else? Maybe it’s easier and clearer to just write “Ref.31 indicated that anomalous anticyclonic circulation over Greenland and the Arctic Ocean in summertime may contribute...”?

Reply: Changed as suggested. (*Please see L52-53*)

L60: “essential” → essence

Reply: Changed. (*Please see L63*)

L80: I’d write “correlation coefficients” instead of “temporal expansion coefficients”, to avoid any potential confusion.

Reply: Changed. (*Please see L86*)

L102-103: “may act as a precursor to drive the eastern Arctic” → “may act as a precursor driving eastern Arctic ...”

Reply: Changed. (*Please see L117-118*)

L130-131: It’s too early in the paper to assert that everything shown is causally responding to the APO (“as a response to the aforementioned ... APO ...”). At this point there are still outstanding questions about causality which the authors have not commented on. Please rephrase to something more cautious, e.g. “This suggests that the thermodynamic conditions associated with the positive summer APO are conducive to a decline in eastern Arctic sea ice in the autumn”, or similar.

Reply: Rephrase as suggested. (*Please see L146-147*)

L199: “concerned” → “concerning”

Reply: Changed. (*Please see L221*)

L201: “makes an advance to this understanding” → “advances this understanding”

Reply: Changed. (*Please see L223*)

L202: I suggest removing the “known as”. It’s not that the APO is known as a mode of variability, it’s that it *is* a mode of variability.

Reply: Deleted.

L205: North Atlantic SSTs should be plural?

Reply: Changed. (*Please see L227*)

L206: Again, SST should be plural (warmer SSTs). Please check other instances in the paper for this.

Reply: We have checked it throughout the manuscript and changed “SST” to “SSTs”.

L216: Should be an “a” between reproduce and significant.

Reply: Added. (*Please see L235*)

L223: “produces” → “shows”

Reply: Changed. (*Please see L242*)

L244: The use of the word ‘contradicts’ seems a bit odd to me. I think ‘counteracts’ would be better.

Reply: It has been changed to “counteract”. (*Please see L268*)

L245: “in the projection of Arctic sea ice” → “in projections of Arctic sea ice”.

Reply: Changed. (*Please see L270*)

L265-266: What exactly is the selection criteria? Is it that only members showing statistically significant correlations are chosen? If so, with respect to what p-value and what test (see major comment!) Or correlations close in magnitude to observations? If

so, what exact threshold? Also, how many members are actually selected in this way? The answer to this would help give an idea of how robust the APO-ice link really is in CESM (and by extension other models).

Reply: These issues have been clarified in the subsection “Community Earth System Model Large Ensemble (CESM-LE) Simulations”. (*Please see L296-304*)

Comment on Figures: I found it very difficult to see the landmasses in Figures 2a and 2b. Is it possible for the authors to make the land more visible? Also, in Figures 4c and 4d, can the authors add information (in caption or as text within the subplots) about when it's the APO leading and lagging?

Reply: The land contours have been thickened in Figures 2a and 2b. The information about when it's the APO leading and lagging in Figures 4c and 4d has been added in the caption.

Response to the Reviewer #3

The aim of the manuscript is to analyse the role of a specific atmospheric variability pattern in summer, namely the Asian-Pacific Oscillation (APO), as a potential driver of autumn sea ice variability in the eastern Arctic. The authors proposed a chain of dynamical processes linking variations in the APO to autumn sea ice loss in the eastern Arctic, whereby the recent trend of the summer APO towards its positive phase may have contributed to accelerated autumn sea ice loss in the eastern Arctic in autumn.

This study continues a series of papers by the lead author dealing with the APO and its relationship with climate variability over the North Pacific or North Atlantic, respectively.

This study contributes to the large body of literature investigating the connections between the Arctic and the mid-latitudes. While the majority of papers investigate how the dramatic Arctic changes in autumn and winter, in particular sea ice retreat and increases in near-surface temperature, may trigger tropospheric and stratospheric pathways from the Arctic to the mid-latitudes, this study contributes to the less studied link between changes in the mid-latitude circulation and the Arctic climate system, in particular Arctic sea ice retreat in autumn (which in turn may trigger the above-mentioned Arctic-mid-latitude linkages in autumn and winter). This study is therefore timely and relevant.

The manuscript is clear and mostly easy to follow, but in my opinion several aspects of the proposed chain of dynamical processes linking variations in the APO to autumn sea ice loss in the eastern Arctic lack evidence, in particular the link from the anomalous circulation over the Asian-North Pacific sector to the anomalous circulation in the mid-latitudes of the North Atlantic. Because this is a key link in connecting SUMMER APO to AUTUMN sea-ice loss, in my view it is necessary to provide more evidence in this regard.

In addition, the description of the methods needs to be improved to ensure that the reader can understand all the analyses and hence the subsequent conclusions. This explicitly includes the analysis of the modelling results, in particular the choice of the members for the sub-ensemble.

At this stage, the submitted manuscript needs careful and major revision.

Reply: Positive comments and helpful suggestion from the reviewer are highly appreciated. We have carefully considered all the comments and made substantial revisions. The revisions in the manuscript are highlighted by red color.

Major comments:

(1) My main concern is the missing evidence for the link from the anomalous circulation over the Asian-North Pacific sector to the anomalous circulation in the mid-latitudes of the North Atlantic. This is critical, because this is a key link in connecting SUMMER APO to AUTUMN sea-ice loss. In my opinion, Fig. 3 alone does not show the real extent of the suggested wave train. It is not clear, which centers of action are really connected via wave activity fluxes, nor the start and end of the wave train. In the referenced earlier study (Fig. 9a in Zhou and Wang, JGR, 2015), wave activity fluxes for the suggested wave train are shown, but in my view, the wave train ends over eastern North America due to only very weak wave activity fluxes originating from the stream function anomaly over eastern North America towards the mid-latitude North Atlantic. I would like to ask the authors to provide stronger evidence for the suggested link from the anomalous circulation over the Asian-North Pacific sector to the anomalous circulation in the mid-latitudes of the North Atlantic.

Moreover, in L160, the authors give the reference Zhao et al., JC (2012). These authors argue that "...the westward propagation of the positive anomalies results in an increase of tropospheric temperature over the North America–Atlantic region. As seen later from section 4d, this westward propagation may be forced by Asian surface heating anomalies. Accordingly, significant positive anomalies appear in the troposphere over North America and the Atlantic, Thus the tropospheric temperature over North America and the Atlantic is high (low) when Asian tropospheric temperature is high (low), suggesting a consistently varying feature of tropospheric temperature between Asia and the North America–Atlantic sector." (Zhao et al., JC (2012), p. 6600). Here, a different mechanisms is suggested. Could the authors should analyse and discuss these different mechanisms.

Reply: Thanks for the suggestion. According to the comment, we have added summer horizontal wave activity flux and stream function associated with the summer APO (Fig. 3c) as well as relevant description (*Please see L166-170*) in the revised manuscript. Seen from Fig.3, a branch of wave train propagates from the Asian-North

Pacific sector to North America and then to the North Atlantic. We also add the mechanism proposed by Zhao et al. (2012) for discussion in the revised manuscript (*Please see L162-166*).

(2) Parts of the "Data and Methods" section have to be improved by providing necessary information to enable the reader to better understand the analyses performed in the study.

With regard to the simulations: more details have to be given, how those ensemble members have been selected, which simulate an summer APO- autumn SIC relation, how many ensemble members from the 40 members simulate the APO-SIC relation, and how the ensemble patterns shown in supplementary Figs. 6 and 7 have been calculated. What are potential reasons that not all ensemble members reproduce the observed summer APO- autumn SIC relation?

With regard to statistical analyses: Please provide information, how the perturbations (needed for the calculation of the wave activity fluxes) have been calculated, since the definition of the basic flow has an impact on the calculated wave fluxes.

Reply: According to the suggestion, we have added more details in the subsection "Community Earth System Model Large Ensemble (CESM-LE) Simulations" to address the issues concerned by the reviewer (*Please see L296-304*). The description for the calculation of the perturbations has also been added in the revised manuscript (*Please see L328-330*).

(3) The authors have used seasonally (3 months) averaged data for their analyses. How robust are their results when using data with higher temporal resolution (monthly means or means over pentads)?

Reply: Thanks for the comment. This study focuses on the seasonally averaged data in order to filter some high frequency signal. However, as a response to the comment, we have calculated the results using the data month by month, although it is beyond the scope of this study. In general, similar results can be obtained. Below (Figs. R2-4) shows the couple of August UTT with the SIC in September, October, and November as an example.

Fig. R2 a–b Spatial patterns of the first leading MCA mode for **a** August UTT over the Asian-Pacific sector and **b** September SIC in the eastern Arctic over the period 1950–2019. Areas significant above the 90% confidence level are dotted. **c** Time series of the UTT (red line) and SIC (blue line) patterns shown in **a** and **b**, respectively.

Fig. R3 Same as Fig. R2, but for August UTT and October SIC

Fig. R4 Same as Fig. R2, but for August UTT and November SIC

Minor comments:

(1) General comment for figures 2, 3, 5, S4, S7: I strongly recommend to show the same area in all these figures for better comparability, and in particular to extend the area shown in fig. 5 to the west (to allow a better identification of the begin and end of wave trains).

Reply: According to the comment, we have adjusted the areas. We extend the longitudes in fig. 5 and S7 (now as S9) to $-180^{\circ}\sim 180^{\circ}$, same as that ($0^{\circ}\sim 360^{\circ}$) in S4 (now as S6). For better shows, Fig.5 and S9 is displayed from the Western Hemisphere to the Eastern Hemisphere, while S6 is displayed from the Eastern Hemisphere to the Western Hemisphere. However, to keep figure clear, Figures 2 and 3 are unchanged.

(2) Introduction L32-67: For some statements, newer references should be added, I here give one example: Arctic amplification: Arctic warming nearly four times faster than the globe (Rantanen et al., 2022). Please check newer publications also for other statements.

Rantanen, M., Karpechko, A. Y., Lipponen, A., Nordling, K., Hyvärinen, O., Ruosteenoja, K., ... & Laaksonen, A. (2022). The Arctic has warmed nearly four times

faster than the globe since 1979. *Communications Earth & Environment*, 3(1), 168.

Reply: We have added the newer publications in this part.

(3) L39: What means etc.?

Reply: We change it to “and so on”. (*Please see L39*)

(4) L34-L44: Please give also refernces for the uncertainties and knowledge gaps with regard to the impacts of Arctic changes on the local and remote weather and climate.

Reply: We have clarified this issue in the revised manuscript. (*Please see L44-49*)

(5) L65: "the reanalysis and simulation data" please, be more specific.

Reply: We have specified it in the revised manuscript. (*Please see L68-71*)

(6) L70 and following: I strongly recommend to reformulate the frist sentences of the results section to allow the reader an easier entry into the results section.

Reply: Reformulated. (*Please see L77-81*)

(7) L74-L76: Please give lat-lon information for the Asian-Pacific region in the text.

Reply: We have added the lat-lon information in the revised manuscript. (*Please see L79-80*)

(8) L74-L76: How sensitive are the results of MCA (or EOF) on the chosen area for the Asian-Pacific region? Does the dipole pattern of the APO appear as first mode for a larger region as well?

Reply: We perform the MCA and EOF analyses for a larger region over the Asian-Pacific region (15°-75°N, 60°E-105°W). Similar results can be obtained (please see Figs. R5-6 below).

Fig. R5 a–b Spatial patterns of the first leading MCA mode for **a** summer UTT over the Asian-Pacific sector and **b** autumn SIC in the eastern Arctic over the period 1950–2019. Areas significant above the 90% confidence level are dotted. **c** Time series of the UTT (red line) and SIC (blue line) patterns shown in **a** and **b**, respectively.

Fig. R6 a EOF1 mode of summer UTT ($\times 10^{-2} \text{C}$) over the Asian-Pacific sector for the period 1950–2019. **b** Normalized time series (PC1; bar) of the EOF1 mode from 1950 to 2019.

(9) L98: How are SIC anomalies calculated?

Reply: The SIC anomalies are calculated relative to the climatology of 1950-2019.

(Please see L110-111)

(10) L488/Fig.1: Please indicate the correlation coefficient between the time series in Fig. 1c. Same for Fig. S5c.

Reply: Added in Fig. 1c and Fig. S5c (now as Fig. S7c).

(11) Fig. S1: The shown timeseries for the APO (PC1 of EOF1 mode over Asian-Pacific sector) deviates from that shown in Zhou et al., 2015 over the period 1965-2000. In Zhou et al. (2015), Fig. 3, I do see interannual variability of the index with no preference of a negative phase, whereas the timeseries shown in Fig. S1 does show a prolonged negative phase from ~1965 to ~2000. Please, could you comment on these differences?

Reply: This is due to the difference in the length of time period and months used in this study from that used in Zhou et al. (2015). Note that the data in both studies are detrended. In Zhou et al. (2015), the time period is 1958-2002. After detrending, it mainly reflects interannual variability. In this study, the time period is much longer (1950-2019). After detrending, it reflects both interannual and decadal variability. In fact, the time period in Zhou et al. (2015) lies in the negative phase of this study. If we extract the time period 1965-2000 from the figure of this study, it also only present interannual variability.

REVIEWER COMMENTS

Reviewer #2 (Remarks to the Author):

I thank the authors for addressing my comments, as well as those of the other two reviewers. I think the paper now only requires what I'd consider "minor revisions" in order for me to recommend for publication.

Minor comments:

L108: The authors have previously stated that the ice and atmosphere MCA timeseries enjoy a correlation of 0.62 with each other. For completeness, it would be good if the authors can also add (after L108) what the correlation is between your APO and SIC indices. Because your indices correlate strongly with the MCA timeseries, I expect it's probably close to 0.62 as well, but I think it would be better to just state the correlation anyway here to avoid any confusion.

L178: The authors refer to earlier modelling studies here, and you also referred to these in your response to my major comments. However, the content of these modelling studies was not really made clear in both cases. After looking at Ref. 60 myself, I would strongly urge the authors to explicitly spell out the key result from Ref. 60 here in the manuscript: when Zhao et al. impose the NA SST pattern in a GCM it generates a local response that differs from the APO-like response diagnosed using regression analysis, and furthermore does not trigger an actual APO anomaly in Asia/Pacific; on the other hand, when APO anomalies are imposed, an NA SST/circulation response is generated which looks similar to the diagnosed one. To me, this is a very strong bit of evidence for the dominant role of the APO, as opposed to just North Atlantic SSTs by themselves. I still think the lead-lag analysis is not convincing by itself, but when combined with Ref 60 and your new BMME/WMME ensemble it becomes much more convincing.

For example, you could do it like this: (L175) "This finding suggests that the main physical process of air-sea interaction in the two regions is the APO forcing on the ocean rather than its response to the oceanic forcing. This is consistent with results from the modeling studies 60,61, which show that when an APO anomaly is imposed in a general circulation model, a circulation and SST response is generated in the North Atlantic which resembles that diagnosed by regression against the APO index (Figures 3 and 4). On the other hand, when a NA_sst-like anomaly is imposed, a different local circulation response is observed and no APO-like pressure anomalies are generated in the Asia-Pacific sector."

The authors should feel free to do it some other way or reword my suggestion, but please do include a description of the results of the modelling studies one way or another.

L252: It seems a bit strong to say that persistent warming "cannot be simulated", given that Figure S10 shows that the signal is overall a warming one. I suggest rephrasing to rather say that the persistent warming is substantially reduced and no longer statistically significant ($p < 0.05$).

Figure comments: I appreciate the changes made to the continents in the revised manuscript. However, the colour of continents is currently very inconsistent: sometimes they are grey, sometimes they are white (e.g. Figure 1a vs Figure 1b, and Figure 1b vs Figure 2c). For 850hpa variables regions above this level (Greenland, mountains...) are also shown in grey with white continents. I feel it would greatly improve legibility of your figures if you made this consistent across all figures. My opinion is that the figures where continents are clearly grey are much easier to look at than ones where continents are white, like Figure 2 where there are colours, dots and arrows covering everything and making it hard to see the continents underneath. I recommend making the continents grey in all figures, and then highlighting regions above 850hPa with a different shade of grey (or pure white?). It is possible that doing such changes could make figures like Figure 2 even harder to read, because of the need for a good contrast between continents and black arrows. I therefore won't insist on these

changes, and leave it to the author's discretion to make the final decision. It is primarily Figures 1a and 2 (and corresponding Supp. Figures) that are difficult to look at: it would be good if the authors explore possible ways of improving legibility one way or another.

Comment on statistics: "The Monte Carlo test" is completely ambiguous and doesn't clarify in any way what statistical test has been done. Monte Carlo just means you wrote down an explicit null hypothesis model and sampled from it many times. The key thing is that you need to explicitly state what the null hypothesis was. In this case, you need to explicitly state how you are choosing to model SSTs, ice and atmospheric variables. Please write down in the Methods what the null hypothesis you used is exactly for your Monte Carlo test. If you went with my suggestion, you are modelling SSTs and ice using an AR1 process (you should clarify that this means 'autoregressive process with a lag of 1 year' in your context) and atmospheric variables as white noise.

Reviewer #3 (Remarks to the Author):

I very much appreciated the efforts made by the authors to improve the manuscript, and the very careful and detailed responses to all the reviewer's comments. The manuscript has clearly improved, but still several aspects of the proposed chain of dynamical processes linking variations in the APO to autumn sea ice loss in the eastern Arctic lack convincing explanations and evidence, in particular the link from the anomalous circulation over the Asian-North Pacific sector to the anomalous circulation in the mid-latitudes of the North Atlantic.

Because this is a key link in connecting SUMMER APO to AUTUMN sea-ice loss, in my view it is still necessary to provide better explanations and in-depth discussions of uncertainties in this regard. Therefore, I still see the need for major revision.

Major comments:

(1) Still, my main concern is the missing evidence for the link from the anomalous circulation over the Asian-North Pacific sector to the anomalous circulation in the mid-latitudes of the North Atlantic. This is critical, because this is a key link in connecting SUMMER APO to AUTUMN sea-ice loss.

As far as I understood from the literature there have been two mechanisms suggested:

(a) "the anomalous pattern over the Asian-North Pacific sector is teleconnected with an anomalous anticyclonic circulation in the mid-latitudes of the North Atlantic through a zonal wave train propagating from the Asian North Pacific region to the North Atlantic via North America (Fig. 3c)." (L160-163). I appreciated the inclusion of Fig. 3c, but I am struggling to see anomalous anticyclonic circulation in the mid-latitudes of the North Atlantic. I recognize cyclonic anomalies over the North Atlantic between 40-60N, and anticyclonic anomalies south of it. In addition, the anticyclonic center of the wave train over eastern North America is only weakly connected to the anomalies over the North Atlantic. Please, clarify this and be more precise in the description of the locations of the centers of the wave train.

A similar plot was shown in a previous paper by the lead author (Zhou and Wang, JGR, 2015, Fig. 9a). As far as I understand, this plot should be directly comparable to Fig. 3c. The wave trains visible in these two plots do show strong similarities over the North Pacific and over North America, but in Fig. 9a from Zhou and Wang (2015), the wave train does not extend into the North Atlantic. I would like to ask the authors again (see also major comment 1 from my first review) to discuss these different results, and give arguments, why you consider the weak extension of the wave train into the North Atlantic visible in Fig. 3c as a strong evidence for the link from the anomalous circulation over the Asian-North Pacific sector to the anomalous circulation in the mid-latitudes of the North Atlantic.

(b) In that respect, I also found arguments for another process leading to increased temperatures over the North Atlantic in the literature which I mentioned in my first review. Zhao et al., JC (2012) argue that "...the westward propagation of the positive anomalies results in an increase of tropospheric temperature over the North America-Atlantic region. As seen later from section 4d, this westward propagation may be forced by Asian surface heating anomalies. Accordingly, significant

positive anomalies appear in the troposphere over North America and the Atlantic, Thus the tropospheric temperature over North America and the Atlantic is high (low) when Asian tropospheric temperature is high (low), suggesting a consistently varying feature of tropospheric temperature between Asia and the North America-Atlantic sector." (Zhao et al., JC (2012), p. 6600).

Although the authors included now a statement in response to my comment from the first review, they do not discuss the importance of each of these two different mechanisms (a) and (b) nor they provided any analysis to show the westward propagation of the Eurasian tropospheric temperature anomalies.

Overall, I still do not see enough evidence for the link from the anomalous circulation over the Asian-North Pacific sector to the anomalous circulation in the mid-latitudes of the North Atlantic, which is really key in the proposed chain of processes to explain the connection from SUMMER APO to AUTUMN sea-ice loss.

(2) I have some questions/comments to the results from the model ensemble.

Do all ensemble members show an APO?

What are potential reasons that not all ensemble members reproduce the observed summer APO-autumn SIC relations (I have asked this already in my first review!)

In my understanding, the main difference between the BMME and WMME ensemble is the difference in the lead-lag correlation between APO and North Atlantic SST (persistent North Atlantic warming in the BMME ensemble). Please, comment on possible reasons for those differences. Is it due to differences in the atmospheric wave train in summer (proposed as reason for increased North Atlantic SST) or does the background ocean circulation play a role or are there other reasons? How do you interpret the low sizes of the BMME and WMME ensembles (only about 10% of the ensemble size of the full CESM Large Ensemble)?

Response to the Reviewer #2

General comments

I thank the authors for addressing my comments, as well as those of the other two reviewers. I think the paper now only requires what I'd consider "minor revisions" in order for me to recommend for publication.

Reply: We highly appreciate positive comments and helpful suggestion. We have revised the manuscript according to these comments. The revisions in the manuscript are highlighted by red color.

Minor comments

L108: The authors have previously stated that the ice and atmosphere MCA timeseries enjoy a correlation of 0.62 with each other. For completeness, it would be good if the authors can also add (after L108) what the correlation is between your APO and SIC indices. Because your indices correlate strongly with the MCA timeseries, I expect it's probably close to 0.62 as well, but I think it would be better to just state the correlation anyway here to avoid any confusion.

Reply: As suggested, we have added their correlation coefficient in the revised manuscript (*Please see L108-109*).

L178: The authors refer to earlier modelling studies here, and you also referred to these in your response to my major comments. However, the content of these modelling studies was not really made clear in both cases. After looking at Ref. 60 myself, I would strongly urge the authors to explicitly spell out the key result from Ref. 60 here in the manuscript: when Zhao et al. impose the NA SST pattern in a GCM it generates a local response that differs from the APO-like response diagnosed using regression analysis, and furthermore does not trigger an actual APO anomaly in Asia/Pacific; on the other hand, when APO anomalies are imposed, an NA SST/circulation response is generated which looks similar to the diagnosed one. To me, this is a very strong bit of evidence for the dominant role of the APO, as opposed to just North Atlantic SSTs by themselves. I still think the lead-lag analysis is not

convincing by itself, but when combined with Ref 60 and your new BMME/WMME ensemble it becomes much more convincing.

For example, you could do it like this: (L175) “This finding suggests that the main physical process of air-sea interaction in the two regions is the APO forcing on the ocean rather than its response to the oceanic forcing. This is consistent with results from the modeling studies 60,61, which show that when an APO anomaly is imposed in a general circulation model, a circulation and SST response is generated in the North Atlantic which resembles that diagnosed by regression against the APO index (Figures 3 and 4). On the other hand, when a NA_sst-like anomaly is imposed, a different local circulation response is observed and no APO-like pressure anomalies are generated in the Asia-Pacific sector.”

The authors should feel free to do it some other way or reword my suggestion, but please do include a description of the results of the modelling studies one way or another.

Reply: Many thanks for the comment. According to the suggestion, we have added the description of the results of the modeling studies in the revised manuscript (*Please see L204-206*).

L252: It seems a bit strong to say that persistent warming “cannot be simulated”, given that Figure S10 shows that the signal is overall a warming one. I suggest rephrasing to rather say that the persistent warming is substantially reduced and no longer statistically significant ($p < 0.05$).

Reply: We have rephrased it as suggested (*Please see L280-281*).

Figure comments: I appreciate the changes made to the continents in the revised manuscript. However, the colour of continents is currently very inconsistent: sometimes they are grey, sometimes they are white (e.g. Figure 1a vs Figure 1b, and Figure 1b vs Figure 2c). For 850hpa variables regions above this level (Greenland, mountains...) are also shown in grey with white continents. I feel it would greatly improve legibility of your figures if you made this consistent across all figures. My opinion is that the figures where continents are clearly grey are much easier to look at than ones where continents are white, like Figure 2 where there are colours, dots and

arrows covering everything and making it hard to see the continents underneath. I recommend making the continents grey in all figures, and then highlighting regions above 850hPa with a different shade of grey (or pure white?).

It is possible that doing such changes could make figures like Figure 2 even harder to read, because of the need for a good contrast between continents and black arrows. I therefore won't insist on these changes, and leave it to the author's discretion to make the final decision. It is primarily Figures 1a and 2 (and corresponding Supp. Figures) that are difficult to look at: it would be good if the authors explore possible ways of improving legibility one way or another.

Reply: Thanks for the comment. We have re-plotted figures to improve their legibility. In the revised manuscript, after several tries, we set the colour of continents white for consistency except that regions above 850hPa are highlighted in grey. We also thicken continents and set the colour of arrows purple to make continents underneath clearly visible.

Comment on statistics: "The Monte Carlo test" is completely ambiguous and doesn't clarify in any way what statistical test has been done. Monte Carlo just means you wrote down an explicit null hypothesis model and sampled from it many times. The key thing is that you need to explicitly state what the null hypothesis was. In this case, you need to explicitly state how you are choosing to model SSTs, ice and atmospheric variables. Please write down in the Methods what the null hypothesis you used is exactly for your Monte Carlo test. If you went with my suggestion, you are modelling SSTs and ice using an AR1 process (you should clarify that this means 'autoregressive process with a lag of 1 year' in your context) and atmospheric variables as white noise.

Reply: According to the suggestion, we have added details about the null hypothesis and the calculation process of the Monte Carlo test in the revised manuscript (*Please see L375-383*).

Response to the Reviewer #3

I very much appreciated the efforts made by the authors to improve the manuscript, and the very careful and detailed responses to all the reviewer's comments. The manuscript has clearly improved, but still several aspects of the proposed chain of dynamical processes linking variations in the APO to autumn sea ice loss in the eastern Arctic lack convincing explanations and evidence, in particular the link from the anomalous circulation over the Asian-North Pacific sector to the anomalous circulation in the mid-latitudes of the North Atlantic.

Because this is a key link in connecting SUMMER APO to AUTUMN sea-ice loss, in my view it is still necessary to provide better explanations and in-depth discussions of uncertainties in this regard. Therefore, I still see the need for major revision.

Reply: We highly appreciate positive comments by the reviewer. We have carefully considered the comments in the revised manuscript, and the revisions are highlighted by red color. We believe we have addressed these comments to the extend we can.

Major comments:

(1) Still, my main concern is the missing evidence for the link from the anomalous circulation over the Asian-North Pacific sector to the anomalous circulation in the mid-latitudes of the North Atlantic. This is critical, because this is a key link in connecting SUMMER APO to AUTUMN sea-ice loss.

As far as I understood from the literature there have been two mechanisms suggested:

(a) " the anomalous pattern over the Asian-North Pacific sector is teleconnected with an anomalous anticyclonic circulation in the mid-latitudes of the North Atlantic through a zonal wave train propagating from the Asian North Pacific region to the North Atlantic via North America (Fig. 3c)." (L160-163). I appreciated the inclusion of Fig. 3c, but I am struggling to see anomalous anticyclonic circulation in the mid-latitudes of the North Atlantic. I recognize cyclonic anomalies over the North Atlantic between 40-60N, and anticyclonic anomalies south of it. In addition, the anticyclonic center of the wave train over eastern North America is only weakly connected to the anomalies over the North Atlantic. Please, clarify this and be more precise in the description of the locations of the centers of the wave train.

A similar plot was shown in a previous paper by the lead author (Zhou and Wang, JGR, 2015, Fig. 9a). As far as I understand, this plot should be directly comparable to Fig. 3c. The wave trains visible in these two plots do show strong similarities over the North Pacific and over North America, but in Fig. 9a from Zhou and Wang (2015), the wave train does not extend into the North Atlantic. I would like to ask the authors again (see also major comment 1 from my first review) to discuss these different results, and give arguments, why you consider the weak extension of the wave train into the North Atlantic visible in Fig. 3c as a strong evidence for the link from the anomalous circulation over the Asian-North Pacific sector to the anomalous circulation in the mid-latitudes of the North Atlantic.

(b) In that respect, I also found arguments for another process leading to increased temperatures over the North Atlantic in the literature which I mentioned in my first review. Zhao et al., JC (2012) argue that "...the westward propagation of the positive anomalies results in an increase of tropospheric temperature over the North America-Atlantic region. As seen later from section 4d, this westward propagation may be forced by Asian surface heating anomalies. Accordingly, significant positive anomalies appear in the troposphere over North America and the Atlantic, Thus the tropospheric temperature over North America and the Atlantic is high (low) when Asian tropospheric temperature is high (low), suggesting a consistently varying feature of tropospheric temperature between Asia and the North America-Atlantic sector." (Zhao et al., JC (2012), p. 6600).

Although the authors included now a statement in response to my comment from the first review, they do not discuss the importance of each of these two different mechanisms (a) and (b) nor they provided any analysis to show the westward propagation of the Eurasian tropospheric temperature anomalies.

Overall, I still do not see enough evidence for the link from the anomalous circulation over the Asian-North Pacific sector to the anomalous circulation in the mid-latitudes of the North Atlantic, which is really key in the proposed chain of processes to explain the connection from SUMMER APO to AUTUMN sea-ice loss.

Reply: We have rewritten this part to address the concern of the reviewer (*Please see L162-191*).

(a) We reword the sentences to describe the locations more precisely. "the anomalous pattern over the Asian-North Pacific sector is teleconnected with an anomalous

cyclonic circulation between 40°-60°N and an anomalous anticyclonic circulation south of it in the upper troposphere of the North Atlantic (Fig. 3a). The westerly anomalies in between indicates an intensification of the westerly jet stream over the North Atlantic, which can induce anomalous descending motion on the right-hand side of the exit of jet stream through a secondary circulation due to the effect of geostrophic deviation⁶⁰. To compensate the downward outflow, a mass divergence is introduced, yielding an anticyclonic circulation anomaly in the lower troposphere of the mid-latitude North Atlantic (Fig. 3b)” (Please see L162-169).

We also polish the tone about the wave train and clarify what the reviewer mentioned. That is, “We hypothesize the anomalous patterns in the upper troposphere of the Asian-North Pacific sector and the North Atlantic are linked via a zonal wave train propagating from the Asian-North Pacific region to the North Atlantic via North America, with positive stream function anomalies over East Asia and northeastern North America and negative anomalies over the western coast of North America and the North Atlantic north of 40°N (Fig. 3c). Note that the wave activity over eastern North America is relatively weak” (Please see L170-176). Although the wave activity is relatively weak over eastern North America, the wave propagation from eastern North America toward the North Atlantic is observed if we narrow the region of Fig. 3c in the manuscript to that shown in Fig. R1 below.

Fig. R1 Horizontal wave activity flux (vectors, $\text{m}^2 \text{s}^{-2}$) and stream function (shading, $10^5 \text{ m}^2 \text{ s}^{-1}$) at 300 hpa in association with the normalized summer APO index from 1950-2019. Areas significant above the 90% confidence level are dotted.

The comparison of downstream propagation of wave train in this study with that in Zhou and Wang (JGR, 2015) is added in the revised manuscript. That is, “Such a

downstream propagation of wave train was previously described in Ref.⁵², but the wave activity seems weaker in the North Atlantic compared to Fig. 3c, which may be due to that it used different domains of Asia (10°-40°N, 30°-140°E) and the North Pacific (10°-40°N, 180°-90°W) to measure the APO during May-August” (*Please see L176-180*).

(b) We add the discussion of the physical mechanism with that of Zhao et al. (JC, 2012). Ref.⁶¹ argued that the Asian land heating induces a westward propagation of tropospheric temperatures to the Atlantic, leading to increased tropospheric temperatures over the North America–Atlantic region and an anomalous high in the troposphere of the Atlantic. It suggests a consistently varying feature of tropospheric temperatures between Asia and the North America-Atlantic sector, which differs from the APO-related tropospheric temperatures that show positive values over the extratropics of Eurasia and negative values over the extratropics of the North Pacific, North America, and the North Atlantic⁴⁸. Thus, the above two processes may link to different phenomena although both affect atmospheric circulations over the North Atlantic. However, the relative importance of the upstream and downstream wave propagations on the North Atlantic climate remains an open issue, which needs more studies in the future (*Please see L179-191*).

(2) I have some questions/comments to the results from the model ensemble.

Do all ensemble members show an APO?

What are potential reasons that not all ensemble members reproduce the observed summer APO-autumn SIC relations (I have asked this already in my first review!)

In my understanding, the main difference between the BMME and WMME ensemble is the difference in the lead-lag correlation between APO and North Atlantic SST (persistent North Atlantic warming in the BMME ensemble). Please, comment on possible reasons for those differences. Is it due to differences in the atmospheric wave train in summer (proposed as reason for increased North Atlantic SST) or does the background ocean circulation play a role or are there other reasons? How do you interpret the low sizes of the BMME and WMME ensembles (only about 10% of the ensemble size of the full CESM Large Ensemble)?

Reply: As seen from Fig. R2 below, all ensemble members can generally reproduce the APO structure.

Fig. R2 Regressions of summer UTT (°C) over the Asian-Pacific sector with the normalized APO index for each member over the period 1950–2005. Areas significant above the 90% confidence level are dotted.

The purpose of using large ensemble simulations is to validate the physical process that the North Atlantic SSTs play a role in linking summer APO and autumn SIC in the eastern Arctic. Given that not all the members can reproduce the summer APO-autumn SIC relationship, we select the BMME and WMME ensemble members rather than all members to outline the differences in the simulated physical process. If using all members as an ensemble, the signals may be smoothed. We have clarified this in the revised manuscript (*please see L341-342*).

The discussion regarding possible reasons for the differences between the BMME and WMME in simulating the relationship of the APO with the North Atlantic SST is also added (*please see L284-296*). One candidate reason may be due to different seasonal evolution of the oceanic temperatures associated with the APO. Previous studies showed that the oceanic temperature anomalies extending to deep mixed layer are essential to the seasonal persistence of SST anomalies via advection and mixing processes⁶⁸⁻⁷⁰. In the BMME simulation (Supplementary Fig. 11a), the APO-related oceanic warming in the North Atlantic extends from the sea surface to a depth of 150 meters in summer. This deep sub-surface warming lasts from summer to autumn, conducive to the maintenance of SST warming. In contrast, for the WMME simulation (Supplementary Fig. 11b), a shallow oceanic warming which only extends to a 50-meter depth is modeled in summer, failing to yield SST warming in autumn. Certainly, this issue needs further in-depth investigation in the future.

REVIEWERS' COMMENTS

Reviewer #2 (Remarks to the Author):

I thank the authors for their efforts, especially with regards to streamlining the figures. I am now recommending publication with no further revisions.

Reviewer #3 (Remarks to the Author):

I very much appreciated the efforts made by the authors to improve the manuscript, and the very careful and detailed responses to my comments from the last review round. In particular I value the improved description and explanation of the link from the anomalous circulation over the Asian-North Pacific sector to the anomalous circulation in the mid-latitudes of the North Atlantic. In my opinion the manuscript is now ready for publication.

Response to the Reviewer #2

I thank the authors for their efforts, especially with regards to streamlining the figures.

I am now recommending publication with no further revisions.

Reply: Positive comments are highly appreciated.

Response to the Reviewer #3

I very much appreciated the efforts made by the authors to improve the manuscript, and the very careful and detailed responses to my comments from the last review round. In particular I value the improved description and explanation of the link from the anomalous circulation over the Asian-North Pacific sector to the anomalous circulation in the mid-latitudes of the North Atlantic.

In my opinion the manuscript is now ready for publication.

Reply: Positive comments are highly appreciated.